

# Crustal structure of southeast Australia from teleseismic receiver functions

Mohammed Bello[1,2], David G. Cornwell[1], Nicholas Rawlinson[3], Anya M. Reading[4], Othaniel K. Likkason[2]

[1]Department Geology & Geophysics, University of Aberdeen, Aberdeen, UK
[2]Department of Physics, Abubakar Tafawa Balewa University, Bauchi, Nigeria
[3]Department of Earth Sciences, University of Cambridge, UK
[4]School of Natural Sciences (Physics), University of Tasmania, Australia

*Correspondence to*: Mohammed Bello (mbazare13@yahoo.com)

**Abstract.** In an effort to improve our understanding of southeast Australia's enigmatic tectonic evolution, we analyse teleseismic earthquakes recorded by 24 temporary and 8 permanent broadband stations using the receiver function method. Crustal thickness, bulk seismic velocity and internal crustal structure of the southern Tasmanides – an assemblage of Palaeozoic accretionary orogens that occupy eastern Australia – are constrained by our new results which point to: (1) a 39.0 ± 0.5 km thick crust, a relatively high Poisson's ratio (0.262 ± 0.014) and a broad (>10 km) crust-mantle transition beneath the Lachlan Fold Belt. This is interpreted to represent magmatic underplating of mafic materials at the base of the crust; (2) a complex crustal structure beneath VanDieland, a postulated Precambrian continental fragment embedded in the southernmost Tasmanides, where the crust thickens (37.5 ± 1.2 km) towards the northern tip of the microcontinent as it enters south central Victoria but thins south into Bass Strait (30.5 ± 2.1 km), before once again becoming thicker beneath western Tasmania (33.5 ± 1.9 km). The thinner crust beneath Bass Strait can be attributed to lithospheric stretching that resulted from the break-up of Antarctica and Australia and the opening of the Tasman Sea; (3) stations located in the East Tasmania Terrane and eastern Bass Strait (ETT+EB) collectively indicate crust of uniform thickness (~33 km) and a slightly broad Moho transition that reflect a possible underplating event associated with a Palaeozoic subduction system. The relative uniformity of $V_p/V_s$ and Poisson's ratio in VanDieland – suggesting uniformity in composition – could be used in support of the VanDieland microcontinental model that explains the tectonic evolution of southeast Australia.

Keywords: receiver functions, crustal structure, VanDieland, Bass Strait, SE Australia

## 1 Introduction

The Phanerozoic Tasmanides (Collins and Vernon, 1994; Coney, 1995; Coney et al., 1990) comprise the eastern one-third of the Australian continent and through the process of subduction accretion were juxtaposed against the eastern flank of the Precambrian shield region of Australia beginning in the Late Neoproterozoic and Early Palaeozoic (Foster and Gray, 2000; Glen, 2005; Glen et al., 2009; Moresi et al., 2014) (Fig. 1). Persistent sources of debate that impede a more complete understanding of the geology of the Tasmanides include (1) the geological link between Tasmania – an island state in southeast Australia – and mainland Australia, which are separated by the waters of Bass Strait; and (2) the presence and locations of continental fragments from Rodinian remnants that are entrained within the accretionary orogens. Furthermore, the lateral boundaries between individual tectonic blocks and their crustal structure are often not well defined. To date, few constraints



on crustal thickness and seismic velocity structure have been available for regions such as Bass Strait.
Constraints on the Moho transition, crustal thickness and velocity structure beneath Bass Strait derived from
receiver functions (RFs) can therefore provide fresh insight into the nature and evolution of the Tasmanides.
Previous estimates of crustal thickness and structure beneath southeastern Australia have been obtained from
deep seismic reflection transects, wide-angle seismic data, topography and gravity anomalies (e.g. Collins,
1991; Collins et al., 2003; Drummond et al., 2006; Kennett et al., 2011). Earlier RF studies in southeast
Australia (Shibutani et al., 1996; Clitheroe et al., 2000; Tkalčić et al., 2011; Fontaine et al., 2013a,b) suggested
the presence of complex lateral velocity variations in the mid-lower crust that probably reflect the interaction of
igneous underplating, associated thinning of the lithosphere, recent hotspot volcanism and uplift. Furthermore,
the intermediate to high crustal $V_p/V_S$ ratio of 1.70–1.78 in this region (Fontaine et al., 2013a), relative to ak135
continental crust where $V_p/V_S$ is ~1.68, may indicate a mafic composition that includes mafic granulite rocks,
granite-gneiss and biotite gneiss. Body- and surface-wave tomography (Fishwick and Rawlinson, 2012;
Rawlinson et al., 2015) reveal P and S wave velocity anomalies in the uppermost mantle beneath Bass Strait and
the Lachlan Fold Belt. Ambient noise surface wave tomography (Bodin et al., 2012b; Young et al., 2012; Pilia
et al., 2015b, 2016; Crowder et al., 2019) of the southern Tasmanides reveals significant crustal complexity, but
is unable to constrain crustal thickness or the nature of the Moho transition.
The goal of this paper is to provide fresh insight into the crust and Moho structure beneath the southern
Tasmanides using $P$-wave RFs, explain the origin of the lateral heterogeneities that are observed and explore the
geological relationship between the different tectonic units that constitute the southern Tasmanides, thereby
facilitating a better grasp of the region's tectonic history.

## 2 Geological setting

The Palaeozoic-Mesozoic Tasmanides of eastern Australia form part of one of the most extensive accretionary
orogens in existence and evolved from interaction between the East Gondwana margin and the Proto-Pacific
Ocean. The tectonic evolution of the Tasmanides is complex and large-scale reconstructions have proven
difficult. This is evident from the variety of models that have been suggested to explain how the region formed
(Foster and Gray, 2000; Spaggiari et al., 2003; Teasdale et al., 2003; Spaggiari et al., 2004; Boger and Miller,
2004; Glen, 2005; Cawood, 2005; Glen et al., 2009; Cayley, 2011; Gibson et al., 2011; Moresi et al., 2014; Pilia
et al., 2015a,b). Particular challenges arise from multiple subduction events, multiple phases of metamorphism,
entrainment of exotic continental blocks, the formation of large oroclines, recent intraplate volcanism and
subsequent events, including the separation of Antarctica and Australia and the formation of the Tasman Sea.
These challenges are compounded by the presence of widespread sedimentary sequences that hinder direct
access to basement rocks (Fig. 1).
The Tasmanides consist of four orogenic belts, namely the Delamerian, Lachlan, Thomson and New England
Orogens. The Delamerian Orogen - located in the south - is the oldest part of the Tasmanides and has a
southward extension across Bass Strait from Victoria into western Tasmania, where it is commonly referred to
as the Tyennan Orogen (Berry et al., 2008). Between about 514 and 490 Ma, the Precambrian and Early
Cambrian rocks that constitute the Delamerian Orogen were subjected to contractional orogenic event along the





margin of East Gondwana (Foden et al., 2006). Subsequently, the Lachlan Orogen formed in the east, which
contains rocks that vary in age from Ordovician to Carboniferous (Glen, 2005). Gray and Foster (2004) argued
for a tectonic model for the Lachlan Orogen that involved interaction of a volcanic arc, oceanic microplates,
several turbidite thrust systems and three distinct subduction zones. Each subduction zones is linked to the
formation of distinct tectonic terrain: the Stawell-Bendigo zone, Tabbarebbera zone and Narooma accretionary
complex. The limited rock exposure in the Tasmanides as a whole has made direct observation of the Lachlan
Orogen difficult; this is attributed to a large swath of Mesozoic-Cenozoic sedimentary cover and more recent
Quaternary volcanics which obscure a large portion of the underlying Palaeozoic terrane. However, the Lachlan
Orogen contain belts of Cambrian rocks in Victoria and New South Wales that are similar in age to the
Delamerian Orogen (Gray and Foster, 2004).
The presence of Precambrian outcrops in Tasmania and the relative lack of similar age rocks in adjacent
mainland Australia has led to different models which attempted to explain the existence of Proterozoic
Tasmania. For instance, Li et al. (1997) suggested that western Tasmania may be a remnant of a continental
fragment set adrift by Rodianian break-up, whereas Calvert and Walter (2000) proposed that King Island, along
with western Tasmania, rifted away from the Australian craton around ~600 Ma (Fig. 1). Other researchers have
developed scenarios in which the island of Tasmania was present as a separate microcontinental block that was
positioned outboard of the eastern margin of Gondwana before re-attaching at the commencement of the
Palaeozoic (Berry et al., 2008).
A popular model that attempts to reconcile the geology observed in Tasmania and adjacent mainland Australia
is that of Cayley (2011). This model proposes that central Victoria and western Tasmania formed a
microcontinental block called "VanDieland" that fused with East Gondwana at the end of the Cambrian,
possibly terminating the Delamerian Orogeny. VanDieland became entangled in the subduction-accretion
system that built the Palaeozoic orogens that now comprise eastern Australia (Fig. 1). Delineating Precambrian
continental fragments within southeast Australia has proven difficult partly due to more recent sedimentary
cover that obscures large tracts of the Tasmanides. However, if present, they likely have distinctive structural
and seismic velocity characteristics.
**3 Previous geophysical studies**
Imaging techniques previously employed to study crustal structure beneath the Tasmanides include: RF analysis
(e.g. Shibutani et al., 1996; Clitheroe et al., 2000; Chevrot and van der Hilst, 2000; Kennett et al., 2011;
Fontaine et al., 2013a,b), ambient noise tomography (e.g. Saygin and Kennett, 2010; Bodin et al., 2012b; Young
et al., 2013a,b; Pilia et al., 2015a,b; Crowder et al., 2019), studies based on potential field imaging and
numerical modelling (e.g. Gunn et al., 1997; Morse et al., 2009; Moresi et al., 2014; Moore et al., 2015, 2016),
teleseismic tomography (Rawlinson and Urvoy, 2006; Rawlinson and Kennett, 2008; Rawlinson et al., 2015,
2016; Bello et al., 2019b) and seismic reflection and refraction profiling (e.g. Finlayson et al., 1980; Collins,
1991; Direen et al., 2001; Glen et al., 2002; Finlayson et al., 2002; Drummond et al., 2006; Cayley et al., 2011;
Glen, 2013). The work of Clitheroe et al. (2000) used RFs to map broad-scale crustal thickness and Moho
character across the Australian continent. These findings confirmed the previous work of Drummond and
Collins (1986) and Collins (1991) who used seismic reflection and refraction transects to determine that the



Lachlan Fold Belt has the thickest crust (∼50 km) in eastern Australia. Shibutani et al. (1996) applied a genetic
algorithm inversion, a non-linear global optimisation technique, to determine the lithospheric velocity structure
of southeast Australia from teleseismic RFs. They found that the Moho is shallow (30-36 km) and sharp within
the craton and deep (38-44 km) and transitional beneath the Tasmanides. They suggested that underplating or
intrusion of mantle material may have thickened the crust and produced a less distinct contrast across the Moho.
A more recent study by Fontaine et al. (2013a) employed *H-κ* stacking and non-linear RF inversion to
investigate crustal thickness, shear wave velocity structure, as well as dipping and anisotropy of the crustal
layers. Their results also indicated a thick crust (∼48 km) and an intermediate (2-9 km) crust-mantle transition
beneath the Lachlan Fold Belt zone which could be attributed to underplating beneath the crust and/or high
concentrations of mafic rocks in the mid-lower crust. Their results also showed a dipping Moho together with
crustal anisotropy in the vicinity of three seismic stations (YNG, CNB and CAN). In our new work, we have a
much increased data coverage of the study area (southern Tasmanides); this allows us to resolve new features,
and further investigate the presence of structures that have been suggested by previous studies.
Over the last decade, ambient noise tomography has become popular tool for studying the structure of the
Australian crust. Saygin and Kennett (2010) produced the first group velocity maps of the Australian continent
from Rayleigh wave group velocity dispersion in the period range 5.0–12.5 seconds. Limited spatial resolution
(∼ 2° x 2°) in our study region means that this model is only able to represent the structure beneath Bass Strait
as a broad, low velocity anomaly. However, the group velocities exhibit a good correlation with known basins
and cratons. Subsequent studies using denser arrays covering southeast mainland Australia (Arroucau et al.,
2010) and northern Tasmania (Young et al., 2011) show good correlations between group/phase velocity maps
and sedimentary and basement terrane boundaries. In order to account for uneven data distribution, Bodin et al.
(2012b) used a Bayesian transdimensional inversion scheme to generate group velocity maps that span the
Australian continent from multi-scale ambient noise datasets. However, in our study area their model is of low
resolution due to the limited station coverage and hence few details on crustal structure can be inferred. Bodin et
al. (2012a) subsequently applied Bayesian statistics to reconstruct the Moho depth of Australia using a variety
of seismic datasets, which gave an approximate Moho depth of ∼30 km beneath Bass Strait.
Potential field data have also been exploited to study the formation and structure of the Tasmanides. Gunn et al.
(1997) integrated potential field data (magnetic and gravity), seismic reflection data, outcrop geology and well
information to study the crustal structure of the Australian continent. Their study found that the occurrence of
tensional stress oriented NE-SW along basement structures in the Bass Basin is able to explain the formation of
the three major sedimentary basins that overlie dense mafic material, which in turn was formed by mantle
decompression processes associated with crustal stretching. From the interpretation of new aeromagnetic data,
Morse et al. (2009) delineated the architecture of the Bass Strait basins and their supporting basement structure.
Subsequent studies by Moore et al. (2015, 2016) used gravity, magnetic, seismic reflection and outcrop data to
support the hypothesis of a VanDieland microcontinent. Their study showed that VanDieland comprises seven
distinct microcontinental ribbon terranes that appear to have amalgamated by the Late Cambrian with major
faults and suture zones bonding these ribbon terranes together.
While the last few decades has seen important advances and insights made into our understanding of the





southern Tasmanides, there still remains limited data on the deep crustal structure beneath Bass Strait, which is
our region of interest. It is therefore timely that can exploit, using the RF technique, teleseismic data recorded
by a collection of temporary and permanent seismic stations in the region to study the structure of the crust,
Moho and uppermost mantle beneath mainland Australia, Bass Strait and Tasmania.
**4 Data**
A collaboration involving five organisations (University of Tasmania, Australian National University, Mineral
Resources Tasmania, the Geological Survey of Victoria and FROGTECH) deployed the temporary Bass seismic
array from May 2011 to April 2013. It consisted of 24 broadband, three-component seismic stations that
spanned northern Tasmania, a selection of islands in Bass Strait and southern Victoria. The instruments used
were 23 Güralp 40T and one Güralp 3ESP sensors coupled to Earth Data PR6-24 data loggers. The permanent
stations consist of eight broadband sensors managed by IRIS, GEOSCOPE and the Australian National Seismic
Network (ANSN). The distribution of all 32 seismic stations that are used in this study is plotted in Fig. 2.
Earthquakes with magnitudes $m_b > 5.5$ at epicentral distances between $30°$ and $90°$ comprise the seismic
sources used in this analysis (Fig. 3). This resulted in an acceptable azimuthal coverage of earthquakes between
the northwest and east of the array, where active convergence of the Australian and Eurasian plate coupled with
westward motion of the Pacific plate has produced extensive subduction zones. To the south and southwest of
the array, the absence of subduction zones in the required epicentral distance range means that there are
significantly fewer events available for analysis from these regions.
**5 Methods**
**5.1 Receiver functions**
The RF technique (Langston, 1979) uses earthquakes at teleseismic distances to enable estimation of Moho
depth and shear wave velocity structure in the neighbourhood of a seismic recorder. If this technique can be
applied to a network of stations with good spatial coverage, it represents an effective way of mapping lateral
variations in Moho depth and crustal structure. The coverage and quality of broadband data available for this
study provides a sound basis on which to examine the crustal structure of the southern Tasmanides.
A recorded teleseismic wavefield at a broadband station can be described by the convolutional model in which
operators that represent the source radiation pattern, path effects, crustal structure below the station and
instrument response are combined to describe the recorded waveform. By using deconvolution to remove the
effects of the source, path and response of the instrument (e.g. Langston, 1979), information on local crustal
structure beneath the station can be extracted from $P$ - to $S$ -wave conversions at discontinuities in seismic
velocity (Owens et al., 1987; Ammon, 1991).
*P*-wave RFs were determined from teleseismic *P*-waveforms using FuncLab software (Eagar and Fouch, 2012;
Porritt and Miller, 2018), following preprocessing using the seismic analysis code (SAC) (Goldstein et al.,
2003). The complete set of 1765 events (Fig. 3) and 32 stations produced 21,671 preliminary RFs. These RFs
were manually picked using the FuncLab trace editor, and by using the clarity of the direct arrivals as an
acceptance criteria, a total of 9,674 RFs were retained for further analysis. The RFs were computed using an





iterative time-domain deconvolution scheme developed by Ligorria and Ammon, 1999 with a 2.5 s Gaussian
filter width. This is performed by deconvolution of the vertical component waveform from the radial and
transverse waveforms with a central frequency of 1 Hz. This frequency was selected on account of significant
source energy detected in the 1 Hz range of teleseismic *P* arrivals, which are sensitive to crustal-scale
anomalies. It also provides a favourable lateral sensitivity with respect to Fresnel zone width (~15 km at Moho
depth) when the conversions from *P* to *S* are mapped as velocity and crustal thickness variations.
**5.2 *H-κ* stacking**
Having obtained reliable *P*-wave RFs, the *H-κ* stacking technique is used to estimate crustal thickness,
Poisson's ratio and bulk $V_p/V_S$. We apply the method of Zhu and Kanamori (2000) to stations where the direct
$P_S$ (Moho *P*-to-*S* conversion) and its multiples are observed. This technique makes use of a grid search to
determine the crustal thickness (*H*) and $V_p/V_S$ (*κ*) values that correspond to the peak amplitude of the stacked
phases. A clear maximum requires a contribution from both the primary phase ($P_S$) and the associated multiples
($P_pP_S$ and $P_pS_S$ +$P_sP_S$). In the absence of multiples, the maximum becomes smeared out due to the inherent
trade-off between crustal thickness (*H*) and average crustal velocity properties (*κ*) (Ammon et al., 1990; Zhu and
Kanamori, 2000). The *H-κ* stacking algorithm reduces the aforementioned ambiguity by summing RF
amplitudes for $P_S$ and its multiples $P_pP_S$ and $P_pS_S$ +$P_sP_S$ at arrival times corresponding to a range of *H* and
$V_p/V_S$ values. In the *H-κ* domain the equation for stacking amplitude

$$s(H,K) = \sum_{j=1}^{N} w_1 r_j(t_1) + w_2 r_j(t_2) - w_3 r_j(t_3) \tag{1}$$

where $r_j(t_j)$; $i = 1,2,3$ are the RF amplitude values at the expected arrival times $t_1, t_2,$ and $t_3$ of the *Ps*, *PpPs*,
*PpSs* +*PsPs* phases respectively for the $j^{th}$ RF, $w_1, w_2, w_3$ are weights based on the signal to noise ratio
($w_1 + w_2 + w_3 = 1$), and *N* is the total number of radial RFs for the station. s(*H*,*κ*) achieves its maximum value
when all three phases stack constructively, thereby producing estimates for *H* and *Vp/Vs* beneath the station. In
this study, the weighting factors used are $w_1 = 0.6, w_2 = 0.3$ and $w_3 = 0.1$ (Zhu and Kanamori, 2000). The *H-κ*
approach requires an estimate of the mean crustal *P*-wave velocity, which is used as an initial value. Based on
the results of a previous seismic refraction study (Drummond and Collins, 1986), we use an average crustal
velocity of *Vp* = 6.65 km/s to obtain our estimates of *H* and *κ* in the study area, noting that *H-κ* stacking results
are much more dependent on *Vp/Vs* than *Vp* (Zhu and Kanamori, 2000). To estimate the uncertainties in the *H-κ*
stacking results, we compute the standard deviation of the *H* and *κ* values at each station.
*H-κ* stacking can also be used to determine Poisson's ratio, which is a useful parameter for inferring the physical
and compositional properties of the crust  (Christensen, 1996) and providing insight into fractures, fluids, and
partial melt (e.g. Mavko, 1980).  The Poisson's ratio *σ* can be determined from *κ* using the equation

$$\sigma = \frac{1}{2}\left(1 - \frac{1}{\kappa^2 - 1}\right) \tag{2}$$

where $\kappa = V_p/V_S$ . While simple to implement, the Zhu and Kanamori (2000) method can suffer from large





uncertainties due to its assumption of a simple flat-laying layer over half-space with constant crustal and upper
mantle properties. Consequently, there are only two search parameters ($H$ and $\kappa$) plus *a priori* information ($V_p$,
weightings) and it does not account for variation with backazimuth. These problems can cause non-unique and
inaccurate estimates, which can lead to potentially misleading interpretations; for instance a low velocity upper
crustal layer can appear as a very shallow Moho in an $H$-$\kappa$ stacking search space diagram. Also, a dipping Moho
and/or anisotropic layers within the crust can contribute to uncertainty.
**5.3 Nonlinear waveform inversion**
In an effort to refine the crustal model, we invert a stack of the radial RFs by adopting the workflow described
by Shibutani et al. (1996). We divide the waveform data (RFs) into four $90°$ quadrants based on the backazimuth
of their incoming energy. The $1^{st}$ quadrant backazimuth range is from $0°$ and $90°$, and an equivalent range in a
clockwise direction defines the consecutive quadrants. The $2^{nd}$ and $3^{rd}$ quadrants (south-eastern and south-
western backazimuths) have very small numbers of RFs. Data from the $1^{st}$ and $4^{th}$ quadrants are of better quality,
with the $1^{st}$ quadrant showing more coherency than the $4^{th}$ quadrant, which is likely due to the orientation of
surrounding tectonic plate boundaries and hence the pattern of *P*-wave energy radiated towards Australia.
Kennett and Furumura (2008) showed that seismic waves arriving in Australia from the northern azimuths
undergo multiple scattering but low intrinsic attenuation due to heterogeneity in the lower crust and mantle; this
tends to produce prolonged high-frequency coda. An important assumption in our inversion is that we neglect
anisotropy and possible Moho dip, which we assume have a second order influence on the waveforms we use to
constrain 1-D models of the crust and upper mantle.
Visual examination of coherency in *P* to *S* conversions allows us to select a subset of RF waveforms for
subsequent stacking. This resulted in groups of mutually coherent waveforms after which a moveout correction
is then applied to remove the kinematic effect of different earthquake distances prior to stacking using a cross-
correlation matrix approach described in Chen et al. (2010) and Tkalčić et al. (2011). Our strict criteria give
reliable RFs at only 6 out of the 32 stations used for this study. An example of some stacked RFs is given in Fig.

242   4.

**5.3.1 Neighborhood algorithm**
We invert RFs for 1-D seismic velocity structure beneath selected seismic stations using the Neighbourhood
Algorithm or NA (Sambridge, 1999a,b) in order to better understand the internal structure of the crust and the
nature of the transition to the upper mantle. NA makes use of Voronoi cells to help construct a searchable
parameter space, with the aim of preferentially sampling regions of low data misfit. In the inversion process, a
Thomson-Haskell matrix method (Thomson, 1950 and Haskell, 1953) was used to calculate a synthetic radial
RF for a given 1-D (layered) structure. During the inversion, as in Shibutani et al. (1996) and Clitheroe et al.
(2000), each model is described by six layers: a layer of sediment, a basement layer, an upper crust, middle crust
and lower crust, and an underlying mantle layer, all of which feature velocity gradients and potentially, velocity
jumps across boundaries. The inversion involves constraining 24 parameters: *Vs* values at the top and bottom of
each layer, layer thickness and the *Vp/Vs* ratio in each layer (Table 1). The inclusion of *Vp/Vs* ratio as an





unknown primarily aims to accommodate the effects of a sediment layer with limited prior constraints
(Bannister et al., 2003). There are two important controlling parameters required by NA: (1) the number of
models produced per iteration ($n_s$); and (2) the number of neighbourhoods re-sampled per iteration ($n_r$). After a
number of trials we chose the maximum number of iterations to be 5500, with $n_s = 13$ and $n_r = 13$ for all
iterations. We employ a chi-squared ($\chi^2$) metric to compute the misfit function, which is a measure of the
inconsistency between the true $\emptyset_i^{obs}(m)$, and predicted, $\emptyset_i^{pre}(m)$ waveforms for a given model ($m$):

$$\chi_v^2(m) = \frac{1}{v} \sum_{i=1}^{N_d} \left( \frac{\emptyset_i^{obs} - \emptyset_i^{pre}}{\sigma_i} \right)^2 \tag{3}$$

where $\sigma_i$ represents the noise standard deviation determined from $\emptyset_i^{obs}$, as explained by Gouveia and Scales
(1998), and $v$ represents the number of degrees of freedom. Using the above stated parameters, the inversion
targets the 1-D structure that produces the best fit between the predicted and observed RF. Figure 7 and 8
present example results of inversions via density plots of the best 1000 data-fitting *S*-wave velocity models
produced by the NA. The optimum data-fitting model is plotted in red.
**6 Results**
**6.1 *H-κ* stacking results for Moho depth and *Vp/Vs* (including Poisson's ratio)**
Maps depicting crustal thicknesses and average *Vp/Vs* in southeast Australia are plotted from the results
obtained at 14 stations (Fig. 6 and 9). At the remaining stations, we could not detect any clear multiples or
Moho conversions in the RFs from any direction. A previous study by Chevrot and van der Hilst (2000) has
noted that this region is devoid of clear multiples. The crustal thickness for all analysed stations in the study
area varies from $30.0 \pm 2.1$ km (BA11) beneath King island in Bass Strait to $39.1 \pm 0.5$ km (CAN) beneath the
Lachlan Fold Belt, and the variation strongly correlates with topography. The associated *Vp/Vs* values range
from $1.65 \pm 0.07$ (BA11) beneath King island to $1.76 \pm 0.04$ (YNG) beneath the Lachlan Fold Belt. Crust of the
order of 30–34 km thickness occurs beneath much of VanDieland. The mountainous region of the Lachlan Fold
Belt has the deepest Moho at $39.1 \pm 0.5$ km (CAN) and a corresponding *Vp/Vs* value of $1.73 \pm 0.02$. Crust that is
~33 km thick lies beneath the East Tasmania Terrane and Eastern Bass Strait (ETT+EB). Table 2 is a summary
of *H-κ* stacking parameters for the analysed stations.
At ~40 km, the crustal thickness beneath the Lachlan Fold Belt is significant, but decreases southward towards
VanDieland (~32.5 km) and southeastward towards the East Tasmania Terrane and Eastern Bass Strait
(ETT+EB) (~33 km). Overall, the Moho becomes shallower from the southern tip of VanDieland (TAU)
towards and into Bass Strait to the north, before becoming deeper once more under the mainland part of the
VanDieland microcontinental block (Fig. 6a). The crustal thickness is more or less uniform beneath the Lachlan
Fold Belt, East Tasmania Terrane and eastern Bass Strait.
The majority of our study region has a low-to-intermediate Poisson's ratio. Poisson's ratio is highest ($0.262 \pm$
$0.014$) in the Lachlan Fold Belt (see Table 2). In VanDieland, the Poisson's ratios generally decrease northward
into Bass Strait from $0.240 \pm 0.019$ (MOO) to $0.210 \pm 0.013$ (BA11) and then increase into mainland Australia





to 0.226 ± 0.017 (TOO). The relatively average to high values in the Lachlan Fold Belt (0.235 ± 0.017 – 0.262 ±
0.014) are in agreement with the presence of a mafic lower crust, as suggested by a number of other studies
(Drummond and Collins, 1986; Shibutani et al., 1996; Clitheroe et al., 2000; Finlayson et al., 2002). The ratios
in the ETT+EB (0.220 ± 0.008 (BA08) – 0.242 ± 0.005 (BA17)) agree with constraints from seismic reflection
and refraction studies and may indicate a felsic to intermediate (average) crustal composition (Finlayson et al.,
2002; Collins et al., 2003).

**6.2 Nonlinear inversion results**

Results of the NA inversion were successfully obtained for a selection of stations, as shown in Table 2. If the
Moho is defined by a gentle velocity gradient, the base of the velocity gradient is used as a proxy for the Moho
depth, as done in previous RF (e.g. Clitheroe et al., 2000; Fontaine et al., 2013a) and seismic refraction
(Collins, 1991; Collins et al., 2003) studies. We also adopt an upper mantle velocity of $Vp$ = 7.6 km/s (i.e. $Vs$ =
4.3-4.4 km/s for $Vp/Vs$ ratios of 1.73-1.77 at the base of the Moho gradient) following Clitheroe et al. (2000)
who used this value for RF studies, and Collins et al. (2003) who used $Vp$ > 7.8 km/s for their summary of both
seismic refraction and RF results; these $Vp$ values are consistent with global Earth models. Therefore, we also
require the $S$-wave velocity to be > 4.4 km/s beneath the Moho. We present the $S$-wave velocity profiles from
the NA inversion for stations CAN, MOO, TOO and YNG in Figs. 7 and 8, together with observed and
predicted RFs. The $S$-wave velocity inversion results of the remaining two stations are included as
supplementary material (see Fig. S.10). In assigning the Moho depth, we consider three criteria to examine the
quality of the inversion result: (1) misfit value ($\chi^2$); (2) the quality of the RF stack (which is based on our
ability to pick the direct and multiple phases); and (3) the visual fit between the synthetic and observed RF.
Models that fail to fit significant arrivals in the observed RF are rejected. Based on these criteria, the inversion
results are classified as:
• Very good: very low $\chi^2$ (typically < 0.4), very good visual fit to direct and multiple phases.
• Good: low $\chi^2$ (typically 0.4-0.8), direct phases clearly visible, multiple phases less clear, and a good
visual fit to all major identifiable phases.
• Poor: medium to high $\chi^2$ (in the range 0.8-1.2), direct phases visible, multiple phases unclear, and
moderate visual fit to some identifiable phases. Looking at the character of the crust-mantle transition,
this study classifies the transition zone as sharp ≤ 2 km, intermediate 2-10 km or broad ≥ 10 km as
initially proposed by Shibutani et al. (1996) and modified by Clitheroe et al. (2000).

**7 Discussion**

For convenience, the seismic stations were separated into three groups (Fig. 2 and Table 2) based on tectonic
settings and the results obtained. Stations YNG, CAN, CNB, MILA and BA13 are located in the Lachlan Fold
Belt; stations BA02, BA11, TAU, MOO and TOO sit above the VanDieland microcontinental block; and
stations BA07, BA08 and BA17 lie in the East Tasmania Terrane and Eastern Bass Strait (ETT+EB). This
discussion focuses on crustal thickness and the nature of the Moho from $H$-$\kappa$ stacking and the nature of the crust



from *Vp/Vs*, Poisson's ratio and the 1-D *S*-wave velocity models.

**7.1 Lateral variation of crustal thickness and nature of the Moho**

The RF analysis clearly reveals the presence of lateral changes in crustal thickness that span mainland Australia
through Bass Strait to Tasmania. The stations located in the Palaeozoic Lachlan Fold Belt reveals a generally
thick crust that ranges from 36.5 ± 4.4 to 39.1 ± 0.5 km. At station CAN, there is a disparity in crustal thickness
obtained by the non-linear inversion method (~49 km) and *H-κ* stacking technique (39.1 ± 0.5 km). The reason
appears to be that the *H-κ* stacking analysis assumes that the crust is a single layer with a velocity jump across
the Moho, whereas the crust-mantle transition is actually gradual; hence it instead targets a shallower boundary
that is not the Moho. Therefore, the deep crustal structure obtained at YNG, CAN and CNB is part of a broad
velocity transition zone from crust to mantle. The crustal thickness and Moho transition zone beneath the
Lachlan Orogen obtained by the nonlinear inversion method is consistent with previous refraction and RF
studies (Shibutani et al., 1996; Clitheroe et al., 2000; Collins et al., 2003; Fontaine et al., 2013a,b). The crustal
thickness variations and lack of a clear Moho at the base of the Lachlan Orogen crust may be a consequence of
mafic magmatic underplating  (e.g. Drummond and Collins, 1986; Shibutani et al., 1996; Clitheroe et al., 2000),
sourced from the ambient convecting mantle. This reinforces the opinion of Glen et al. (2002), who suggested
that the deep Moho underlying the Lachlan Orogen results from magmatic underplating that added a thick
Ordovician mafic layer at the base of the crust coupled with a thick sequence of Ordovician mafic rocks that can
be found in the mid and lower crust. Finlayson et al. (2002) and Glen et al. (2002) also inferred  the presence of
underplating near CNB and CAN from seismic refraction data. Collins (2002) postulated that the underplating
might have occurred in the back-arc region of a subduction zone due to pronounced adiabatic decompression
melting in the asthenosphere. The seismic tomography model of Rawlinson et al. (2010, 2011) exhibits an
increase in *P*-wavespeed at 50 km depth beneath CAN, CNB and YNG and the authors suggest that magmatic
underplating may be the cause of the high velocity anomaly. A recent study by Davies et al. (2015) identified
the longest continental hotspot track in the world (over 2000 km total length), which began in north Queensland
at ~33 Ma, and propagated southward underneath the present day Lachlan Fold Belt and Bass Strait. The
magmatic underplating could therefore be a consequence of the passage of the continent above a mantle
upwelling leading to a more diffuse crust-mantle transition zone. The thickened crust and a transitional Moho
observed in the Lachlan Fold Belt are consistent with the proposed delamination models of Collins and Vernon
350    (1994).

Strong lateral changes in crustal structure and/or composition beneath VanDieland appear to be a reflection of
the region's complex tectonic history (Fig. 6 and 9). The thick crust (37.5 ± 1.2 km) beneath the Selwyn Block –
within the northern margin of VanDieland in southern Victoria – thins (to 30.5 ± 2.1 km) as it enters Bass Strait,
yet in southern Tasmania, at stations TAU and MOO, the crust is thicker (33.5 ± 1.9 km). This is reflected in
both the NA inversion and *H-κ* stacking depth estimates where a sharp Moho is observed beneath this region of
the study area (Fig. 6 and 9). The Moho depth estimates from RFs at stations TAU and MOO (~34 km) is
almost identical (~35 km) to that deduced by Korsch et al. (2002) from a seismic reflection profile adjacent to
the two seismic stations. In contrast, the Bass Strait portion of VanDieland appears to have a relatively thinner
crust (~30 km). This may indicate thinning of the lithosphere associated with lithospheric stretching and or



delamination that resulted from tectonic events that occur post-formation of the Tasmanides (Gaina et al., 1998).
Stations BA07, BA08 and BA17 (ETT+EB) collectively indicate crust of uniform thickness (~33 km, Figures
9a,b). Relative to western Bass Strait, the crust thickens slightly in this part of the study area, which may
suggest underplating associated with a Palaeozoic subduction system (e.g. Drummond and Collins, 1986; Gray
and Foster, 2004). Furthermore, our results support the crustal thickness estimates of Tasmania from refraction
and wide-angle reflection travel time tomography by Rawlinson et al. (2001). They suggested that the
thickening of the crust beneath central northern Tasmania is associated with the suturing of the West and East
Tasmania Terranes during the Middle Devonian Tabberabberan Orogeny. The Moho depths we obtained at
stations TAU, MOO, BA02 and BA11 which are located within their study area show significant overlap in
crustal thickness estimates (Fig. S10 in supplementary material).
In general, our understanding of crustal thicknesses variations are limited by station separation, so it is difficult
to determine whether smooth variations in thickness or step-like transitions explain the observations.
**7.2 Poisson's ratio, $V_p/V_S$ and average crustal composition**
Poisson's ratio, which shares an inverse squared relationship to $Vp/Vs$ (Eq. 2) can constrain chemical
composition and mineralogy more robustly than *P*- or *S*-wave velocity in isolation (Christensen and Fountain,
1975). We observe variations in Poisson's ratio (and hence $Vp/Vs$) across the study region, which we equate
with variations in composition. Studies in mineral physics and field observations show (1) a linear increase in
Poisson's ratio with decreasing $SiO_2$ content in the continental crust and (2) partial melt is depicted by an
elevated Poisson's ratio (>0.30), especially if the anomaly is localised to an intra-crustal layer (Owens and
Zandt, 1997). In terms of $Vp/Vs$, a more felsic ($SiO_2$) composition in the lower crust is represented by a lower
$Vp/Vs$, which reflects removal of an intermediate-mafic zone by delamination, whereas a more mafic lower crust
is depicted by higher $Vp/Vs$ (> 1.75) which may be due to underplated material (Pan and Niu, 2011). However,
lower crustal delamination can also result in decompression melting, which can yield elevated $Vp/Vs$ (He et al.,
2015). We interpret the variation of observed Poisson's ratios (0.210–0.256) in the southern Tasmanides to be a
consequence of compositionally heterogeneous crust and localised partial melt that may likely be sourced from
recent intraplate volcanism (Rawlinson et al., 2017).
Figure 6b shows the distribution of bulk $Vp/Vs$ across the study area. Upon comparison with our Moho depth
results (Fig. 6a and 9a,b), we find that areas of thick crust (Lachlan Fold Belt) do overlap with areas of higher
$Vp/Vs$ (1.70 ± 0.04 − 1.76 ± 0.04). This may strengthen the argument for mafic magmatic underplating sourced
from an ambient convecting mantle (Glen et al., 2002). At MILA, BA13, CAN and CNB, the $Vp/Vs$ values (1.70
± 0.04–1.73 ± 0.06) are consistent with mafic granulite (Christensen and Fountain, 1975) which has been
suggested to occur in the lower crust based on a wide-angle seismic line that cross-cuts the southern region of
the Lachlan Orogen (Finlayson et al., 2002). At station YNG the $Vp/Vs$ value of 1.76 ± 0.04 is consistent with
biotite gneisses deduced from seismic reflection experiments carried out across the Junee-Narromine Volcanic
Belt in the neighborhood of YNG (Direen et al., 2001).
The VanDieland $Vp/Vs$ distribution is rather complex, hence we further divide this block into two separate





groups: (1) West Tasmania Terrane (WTT); (2) and the Selwyn block. In the WTT, stations BA02, TAU, MOO
(see Fig. 2 for the location) have a moderate *Vp/Vs* (1.69 ± 0.02–1.71 ± 0.04). The bulk *Vp/Vs* beneath BA02
(1.69 ± 0.02) supports a dominantly felsic crustal composition, which means that it is unlikely that the WTT has
a mafic lower crust. A felsic crustal composition is at odds with the crustal composition required by the lower
crustal flow model of Drummond and Collins (1986); Gray and Foster (2004). Our *Vp/Vs* measurement from the
permanent GSN station TAU (1.70 ± 0.08), agrees well with *Vp/Vs* value at BA02 which implies a similar
crustal composition. Station MOO adjacent to TAU exhibits a similar *Vp/Vs* value (1.71 ± 0.04) and together
this may indicate that the crust is more or less homogeneous in this region. However, the slight variation in
*Vp/Vs* values between station MOO and TAU may be associated with a slight change in bulk composition and
the effects of heating following juxtaposition of western and eastern Tasmania during the Middle Devonian
Tabberabberan Orogeny.
In Bass Strait and south central Victoria (underlain in part by the Selwyn Block), the abrupt variations in *Vp/Vs*
values across stations BA11 and TOO help to underscore the region's complex tectonic evolution. Very few
reliable *H-κ* stacking parameters were observed in this region: one on King Island (BA11) and the other
adjacent to the NVP in south central Victoria. This is attributed to low signal quality/difficulty in identifying
crustal multiples in this region (Chevrot and van der Hilst, 2000). The presence of a complex and
compositionally variable Selwyn Block beneath the stations (Cayley et al., 2002), and melt-induced heating of
the crust associated with the Quaternary NVP, may also be contributing factors. The *Vp/Vs* value at BA11 (1.65
± 0.07) is the lowest in the study area which may imply a lower crustal delamination in Bass Strait, leaving a
dominantly felsic crust (e.g. He et al., 2015; Bello et al., 2019b).
Station TOO located adjacent to the NVP exhibits a relatively low *Vp/Vs* (1.68 ± 0.04) that implies a more felsic
composition, although mantle upwelling generated by the combined effects of a plume, SDU (shear driven
upwelling) and EDC (edge driven convection) (Rawlinson et al., 2017) would likely yield melts of a mafic
composition, so the low *Vp/Vs* may be caused by something else.
Despite the fact that crustal composition was possibly altered by recent deformational events that resulted from
the break-up between Antarctica and Australia, similar *Vp/Vs* measurements are generally observed from the
southern tip of Victoria through King Island to northwestern Tasmania. This suggests a tectonic relationship
between northwest Tasmania and the Selwyn block and appears to support the presence of a coherent
Precambrian microcontinental block (VanDieland) postulated by several studies in the preceding ~20 years
(Cayley et al., 2002; Cayley, 2011; Moresi et al., 2014; Pilia et al., 2015a).
**8 Conclusions**
We used *H-κ* stacking of teleseismic RFs to determine crustal thickness and *Vp/Vs* ratios; we also generate 1-D
*S*-wave velocity profiles of the crust from 1-D RF inversion in order to investigate the internal crustal velocity
structure beneath the southern Tasmanides. We were able to verify the presence of several crustal structures
imaged by previous studies (Clitheroe et al., 2000; Finlayson et al., 2002; Glen et al., 2002; Reading et al., 2011;
Fontaine et al., 2013a,b) where there is overlap and we have also been able to provide new estimates of crustal
thickness and composition. We have also been able to shed fresh light on the different tectonic blocks that





constitute southeast Australia. The major conclusions are as follows:
•    The thick crust and broad crust-mantle transition beneath the Lachlan Fold Belt may be caused by
magmatic underplating of mafic materials beneath the crust, which is consistent with a relatively high
Poisson's ratio ($0.262 \pm 0.014$). Thicker crust is also to be expected from the elevated topography
beneath the eastern Lachlan Fold Belt.

•    The crustal structure is complex in VanDieland. It thins from the northern tip of the microcontinent
into Bass Strait, yet in southern Tasmania the crust is thicker ($33.5 \pm 1.9$ km) compared to Bass Strait.
This scenario may be attributed to the break-up of Antarctica and Australia and the opening of the
Tasman Sea which formed three failed rift basins that contain thick piles of sedimentary rocks (Gaina
et al., 1998). The thinner crust beneath Bass Strait may indicate that the thinning of the lithosphere is
associated with processes such as delamination and/or stretching of the lithosphere during the break-up
of the two continents.

•    Stations at ETT+EB collectively indicate crust of uniform thickness (~33 km) and an intermediate
Moho transition which possibly reflects underplating associated with a Palaeozoic subduction system.

•    It is clear that the nature of velocity anomalies differ between stations on mainland Australia and
Tasmania. This highlights contrasting lithospheric structure across Bass Strait ($\sim 40^{\circ}$S) with thin
lithosphere to the south and thick lithosphere to the north. This sharp transition of lithospheric
thickness is in agreement with previous results (Clitheroe et al., 2000) and corresponds to changes in
fast $S$-wave polarization directions from primarily northeast-southwest orientations in the north to
nearly northwest-southeast directions in the south (Heintz and Kennett, 2005; Pilia et al., 2016; Bello et
al., 2019a).

Results from this study advance our understanding of the nature and composition of different tectonic blocks
that constitute the geology of the southern Tasmanides. These results will also be important for helping to
understand the results from other comparable seismic imaging studies and the interpretation of tectonic
processes on a wider scale.
**9 Data availability**
Dataset available at 10.6084/m9.figshare.12233723

**10 Author contributions**
M.B. performed the data analysis and wrote the draft manuscript. N.R and D.C. guided the study and assisted in
interpretation. M.B., D.C. and N.R. discussed the results and revised the manuscript. A.R. and O.L. revised the
manuscript and assisted with the interpretation.
**11 Competing Interests:** The authors declare no competing interests.



**12 Acknowledgments**

The work in this paper was performed as part of a PhD study and has been jointly funded by Abubakar Tafawa Balewa University (ATBU), Bauchi, Nigeria and the University of Aberdeen, UK. The authors acknowledge the efforts of staff, students and fieldwork technicians from the Australian National University and University of Tasmania, who deployed the temporary BASS array used in this study. We also thank Qi Li and Armando Arcidiaco for their efforts in BASS data pre-processing and archiving. Australian Research Council Grant LP110100256 supported the BASS deployment. We are grateful to IRIS and Geoscience Australia for providing data from several stations in mainland Australia and Tasmania. Figure 1 was made using Inkscape software (Harrington, et. al., 2005) and Figures 2, 3, 6 and 9 were produced using the Generic Mapping Tools (Wessel et al., 2013).





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

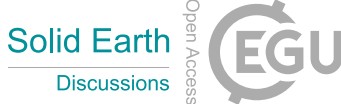







Table 1: Model parameter bounds used in the Neighbourhood Algorithm receiver function inversion. $V_s^{upper}$ and $V_s^{lower}$ represent the $S$-velocity at the top and bottom of a layer respectively. $V_p/V_s$ represents $P$ and $S$ wave velocity ratio within a layer.

| Layer | Thickness (m) | $V_s^{upper}$ (km/s) | $V_s^{lower}$ (km/s) | $V_p/V_s$ |
|---|---|---|---|---|
| Sediment | 0-2 | 0.5-1.5 | 0.5-1.5 | 2.00-3.00 |
| Basement | 0-3 | 1.8-2.8 | 1.8-2.8 | 1.65-2.00 |
| Upper crust | 3-20 | 3.0-3.8 | 3.0-3.9 | 1.65-1.80 |
| Middle crust | 4-20 | 3.4-4.3 | 3.4-4.4 | 1.65-1.80 |
| Lower crust | 5-15 | 3.5-4.8 | 3.6-4.9 | 1.65-1.80 |
| Mantle | 5-20 | 4.0-5.0 | 4.0-5.0 | 1.70-1.90 |




















Table 2: Moho depth, $V_p/V_s$, Poisson's ratio and nature of the Moho for siesmic stations analysed by two different techniques used in this study.

| | Basic station information | | | | Results | | | | |
| Name | No. of RFs | Lon.(°) | Lat.(°) | Moho depth grid search | Moho depth inversion | Quality inversion | Bulk $V_p/V_s$ | Poisson's ratio ($\sigma$) | Nature |
|---|---|---|---|---|---|---|---|---|---|
| **VanDieland** | | | | | | | | | |
| BA02 | 4 | 145.20 | -40.95 | 31.4±2.1 | - | - | 1.69±0.02 | 0.231±0.017 | - |
| BA11 | 12 | 143.98 | -39.64 | 30.5±2.1 | - | - | 1.65±0.07 | 0.210±0.013 | - |
| TAU | 41 | 147.32 | -42.91 | 33.5±1.9 | 33 | poor | 1.70±0.08 | 0.235±0.036 | intermediate |
| MOO | 58 | 147.19 | -42.44 | 33.0±1.2 | 34 | good | 1.71±0.04 | 0.240±0.019 | sharp |
| TOO | 276 | 145.59 | -37.57 | 37.5±1.2 | 35 | good | 1.68±0.04 | 0.226±0.017 | sharp |
| **Lachlan Fold Belt** | | | | | | | | | |
| YNG | 178 | 148.40 | -34.30 | 37.0±1.2 | 48 | good | 1.76±0.04 | 0.262±0.014 | broad |
| CAN | 402 | 149.00 | -35.32 | 39.1±0.5 | 49 | very good | 1.73±0.02 | 0.250±0.008 | broad |
| CNB | 155 | 149.36 | -35.32 | 38.5±1.1 | 46 | good | 1.70±0.04 | 0.235±0.017 | broad |
| MILA | 4 | 149.16 | -37.05 | 37.6±2.1 | - | - | 1.73±0.06 | 0.251±0.023 | - |
| BA13 | 6 | 148.83 | -37.63 | 36.5±4.4 | - | - | 1.72±0.12 | 0.245±0.045 | - |
| **ETT+EB** | | | | | | | | | |
| BA07 | 5 | 148.31 | -40.43 | 32.5±0.1 | - | - | 1.70±0.02 | 0.235±0.001 | - |
| BA08 | 13 | 147.97 | -39.77 | 34.0±1.2 | - | - | 1.67±0.03 | 0.220±0.008 | - |
| BA17 | 5 | 146.33 | -39.04 | 33.2±0.5 | - | - | 1.71±0.02 | 0.242±0.008 | - |




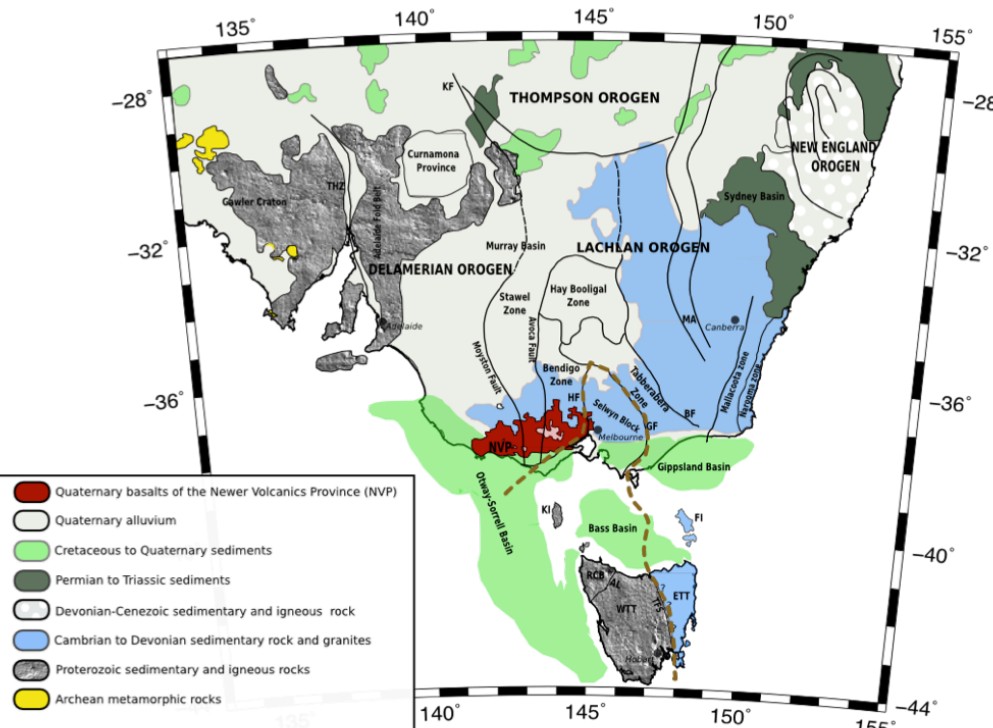

Figure 1: Regional map of southeastern Australia that shows key geological boundaries and the locations of observed or inferred tectonic units (Modified from Bello et al., 2019a). Thick black lines delineate structural boundaries and the thick brown dashed line traces out the boundary of VanDieland. HF = Heathcote Fault; GF = Governor Fault; BF = Bootheragandra Fault; KF = Koonenberry Fault; THZ = Torrens Hinge Zone; MA = Macquarie Arc; NVP = Newer Volcanics Province; KI = King Island and FI = Flinders Island in Bass Strait; ETT = East Tasmania Terrane; WTT = West Tasmania Terrane; TFS = Tamar Fracture System; AL = Arthur Lineament and RCB = Rocky Cape Block. Outcrop boundaries are sourced from Rawlinson et al. 2016.



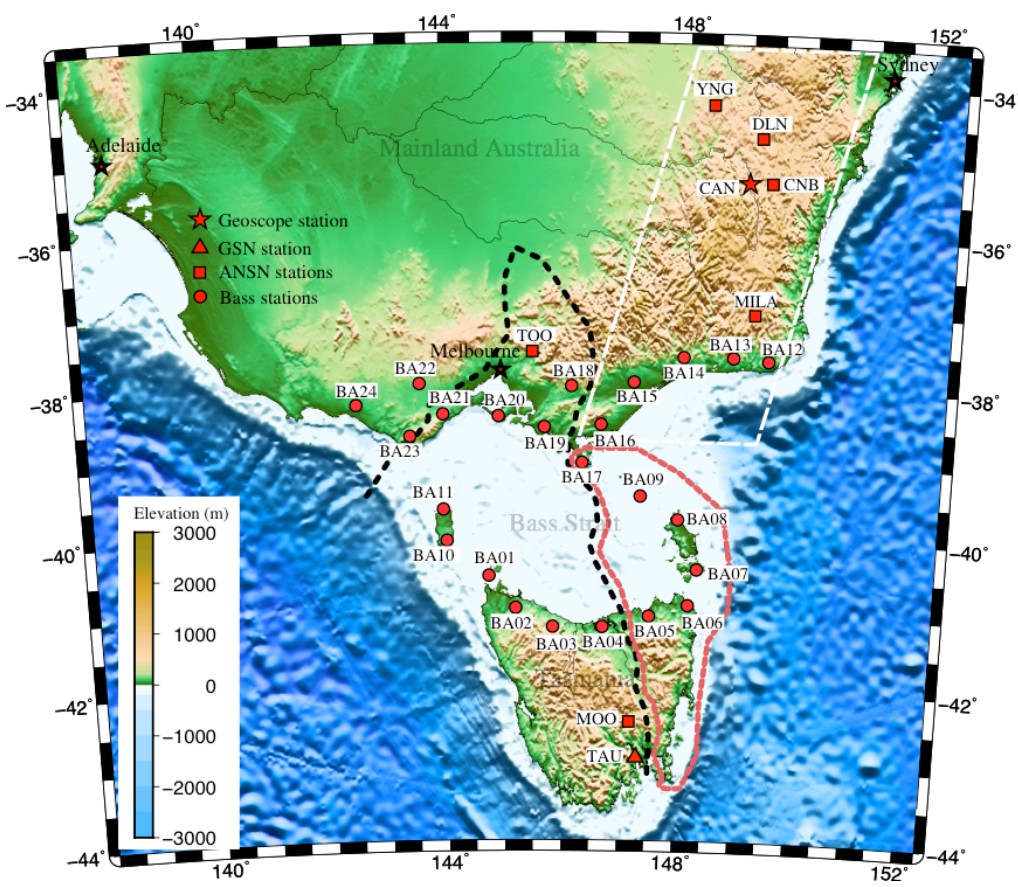

**Figure 2: Location of seismic stations used in this study superimposed on a topographic/bathymetric map of southeast Australia (Modified from Bello et al., 2019a). The boundary of VanDieland is delineated by a thick black dashed line. The boundary of the East Tasmania Terrane and Furneaux Islands is represented by a thick dashed red line, while a thick dashed white line traces out the eastern sector of the Lachlan Fold Belt. Topography/bathymetry is based on the Etopo1 dataset (Amante and Eakins, 2009).**



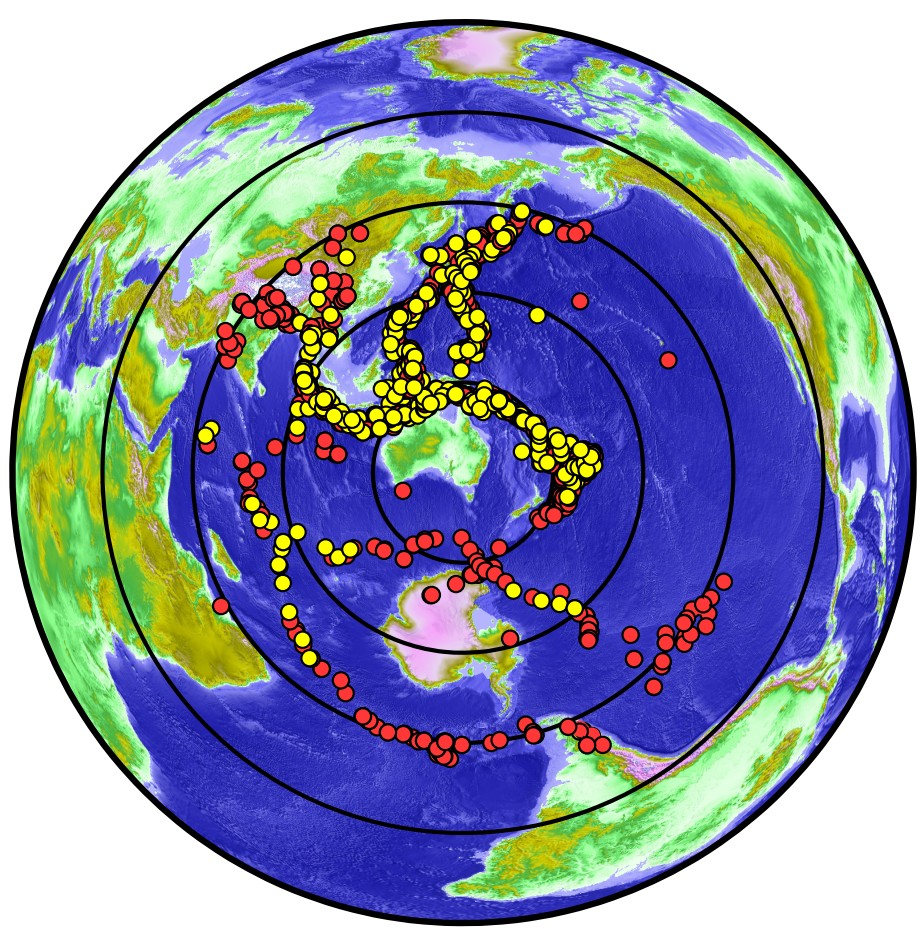

**Figure 3: Distribution of distant earthquakes (teleseisms) used in the study. The locations of events that are**
**ultimately used for RF analysis are denoted by yellow dots. Concentric circles are plotted at 30° intervals from the**
**centre of Bass Strait.** Topography/bathymetry is based on the Etopo1 dataset (Amante and Eakins, 2009).





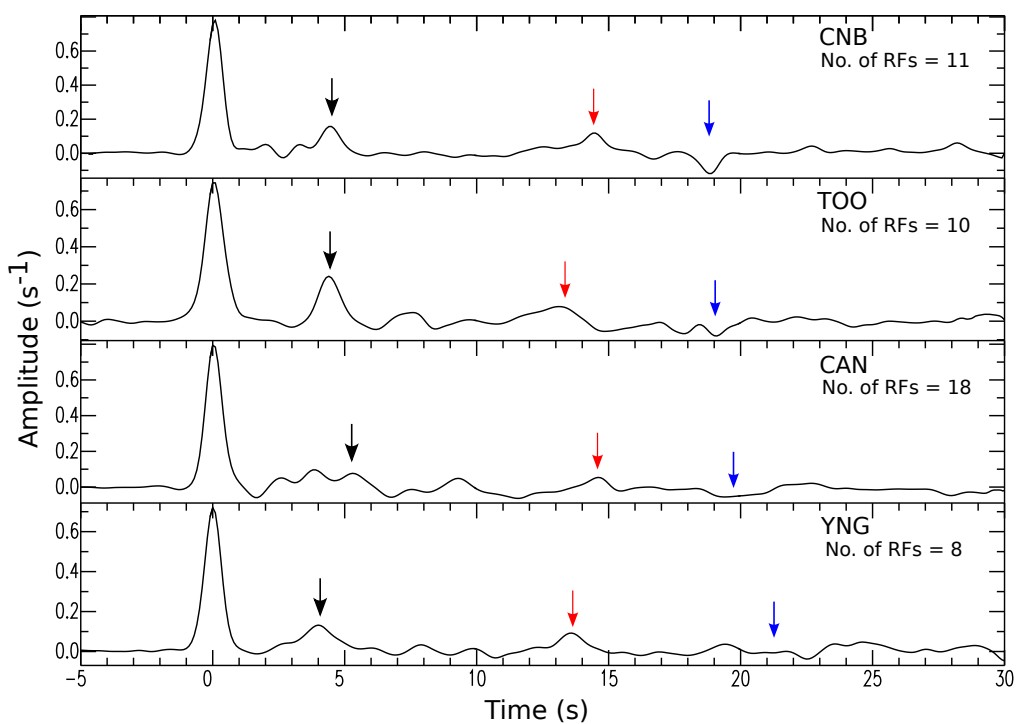

**Figure 4: Stacked receiver functions from Australian National Seismic Network (ANSN) stations TOO, YNG, MOO**
**and GSN station TAU. Small arrows indicate arrival of the *Ps* (black), *PpPs* (red) and *PpPs + PsPs* (blue) phases from**
**the Moho.**





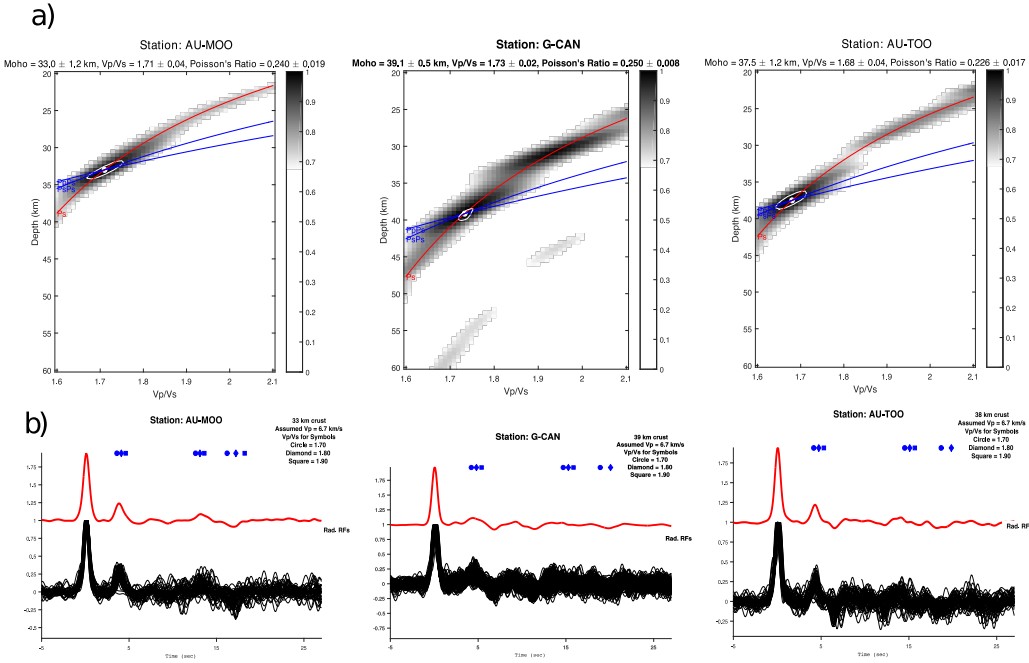

**Figure 5: Results from the *H-κ* stacking analysis for RFs (Zhu and Kanamori, 2000) at stations MOO, CAN and**
**TOO. In each case (a) Normalised amplitudes of the stack over all back-azimuths along the travel time curves**
**corresponding to the $P_S$ and $P_pP_S$ phases. (b) Corresponding stacked receiver function for each station.**












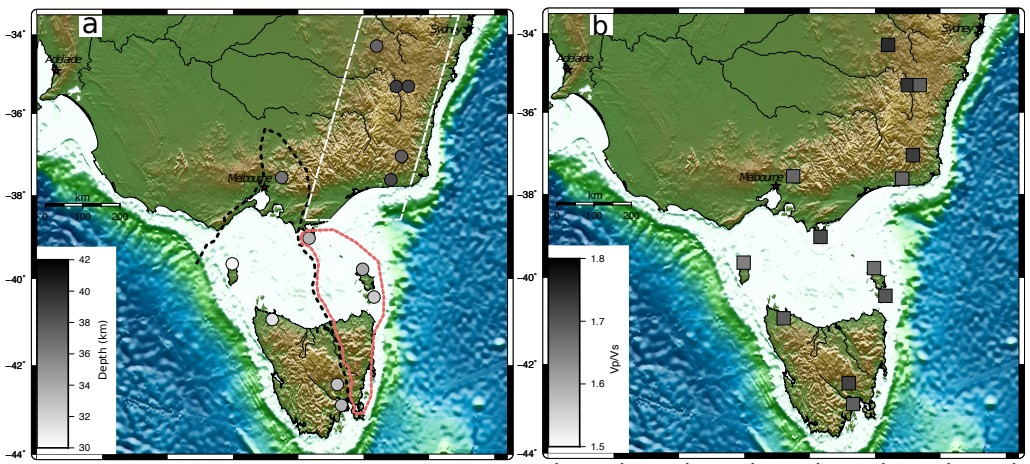

**Figure 6: (a) Variations in crustal thickness and (b) $V_p/V_S$ ratio taken from the linear ($H$-$\kappa$) stacking results (Table 2). Crustal thickness varies between 30.5 ± 0.1 km and 39.1 ± 0.5 km. Thinner crust in Bass Strait can be seen flanked by a relatively thicker crust to the north and south. $V_p/V_S$ ratios vary from 1.65 ± 0.02 to 1.75 ± 0.02. Thick black dashed line denotes the boundary of VanDieland. Thick red dashed line outlines the boundary of East Tasmania Terrane and eastern Bass Strait (ETT+EB). Thick white dashed line highlights the eastern part of the Lachlan Fold Belt. Topography/bathymetry is based on the Etopo1 dataset (Amante and Eakins, 2009).**



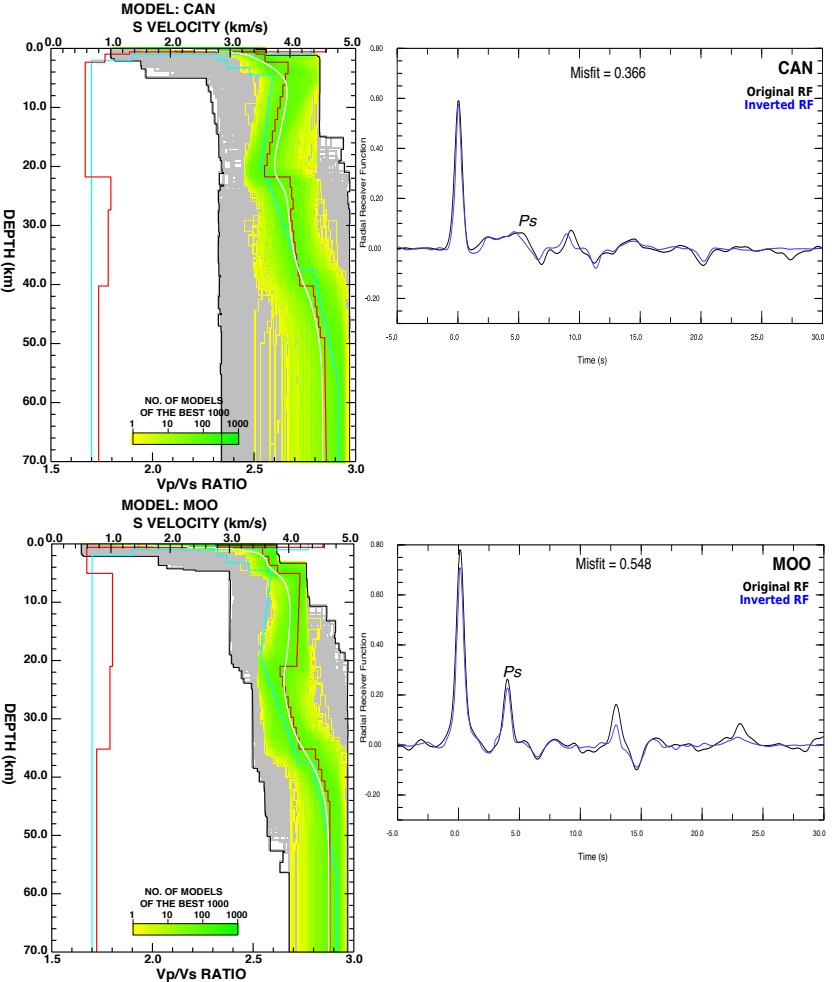


**Figure 7: (Left) Seismic velocity models for CAN and MOO stations obtained from the neighbourhood algorithm**
**(Sambridge 1999a). The grey area indicates all the models searched by the algorithm. The best 1000 models are**
**indicated by the yellow to green colours; the best one (smallest misfit) corresponds to the red line, both for *S*-wave**
**velocity and $V_p/V_s$ ratio and the white line is the average velocity model. (Right) Waveform matches between the**
**observed stacked receiver functions (black) and predictions (blue) based on the best models.**


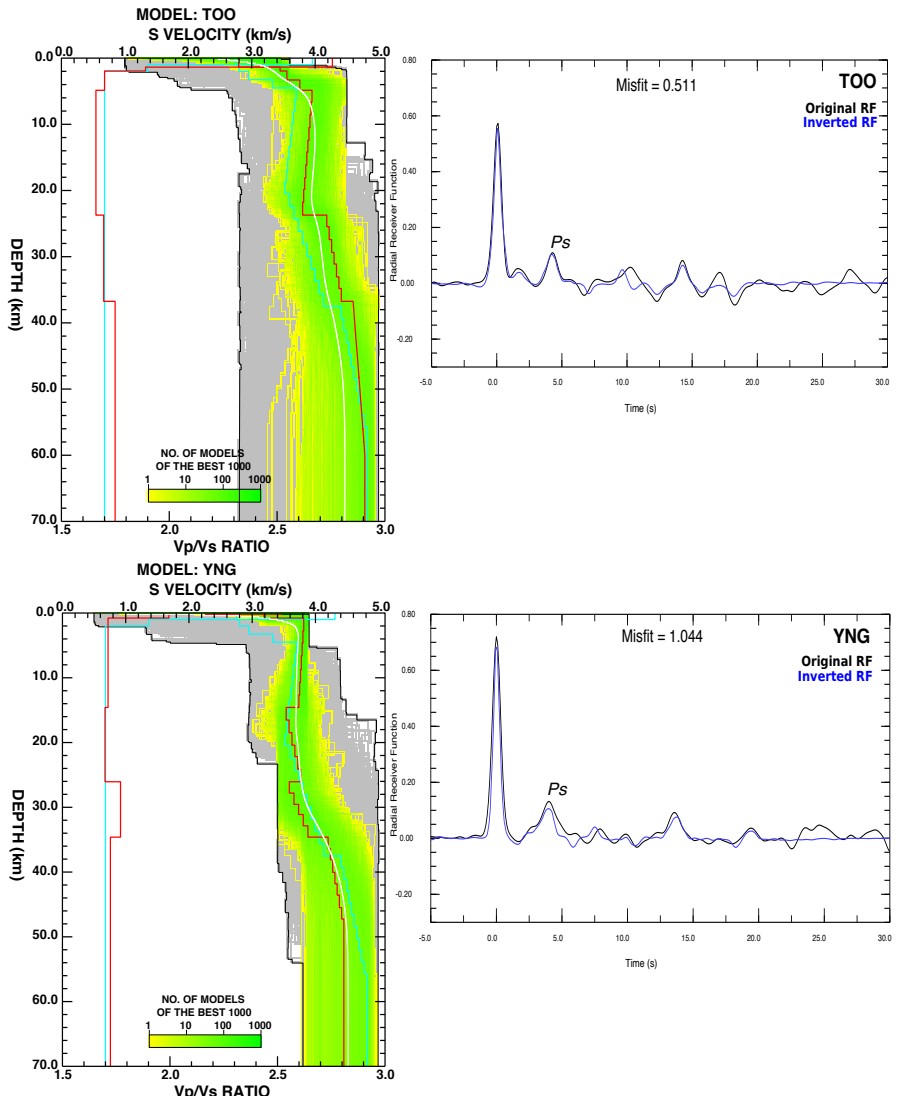

**Figure 8: (Left) Seismic velocity models for stations TOO and YNG obtained from the neighbourhood algorithm.**
**(Right) Comparison between the observed stacked and the predicted receiver functions from the NA inversion. See**
**Figure 7 caption for more details.**





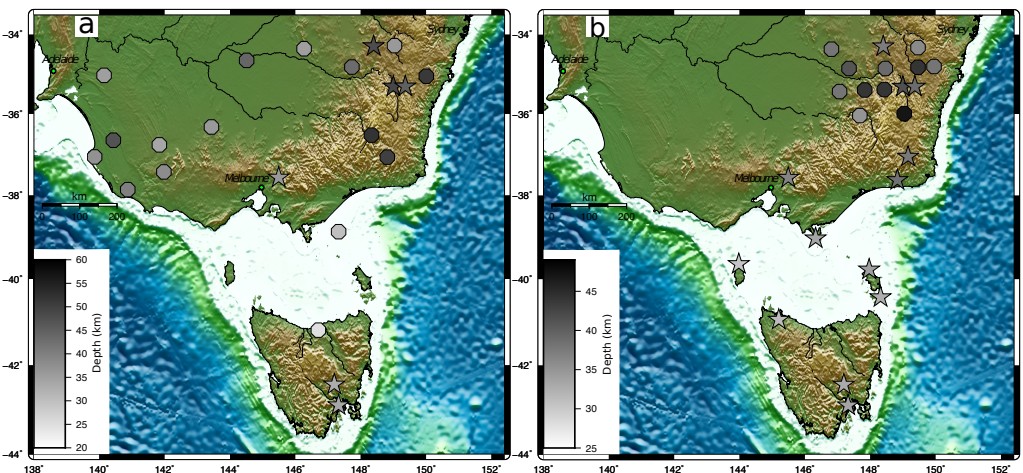

**Figure 9: (a) Map showing crustal thickness variations based on the S-wave velocity inversion results of this study (stars) and previous studies (octagons) (Fontaine et al., 2013; Shibitani, 1996; Collins, 1991) and (b) comparison of crustal thickness variations based on the _H-κ_ grid search results of this study (stars) and previous results from the study of Tkalčić et al. (2012) (octagons). Topography/bathymetry is based on the Etopo1 dataset (Amante and Eakins, 2009).**