# Peer review of "Crustal structure of southeast Australia from teleseismic receiver functions"

_Solid Earth, 2020_

## Referee Comment (RC1) · Anonymous Referee #1 · 28 Jul 2020

In the present manuscript, the authors calculated and evaluated Ps receiver functions at a total of 32 seismic stations in southeast Australia, most of them situated along both sides of Bass Strait (Victoria and Tasmania). They applied H-K stacking and receiver function inversion using the neighborhood algorithm, relatively standard techniques that have already been applied to different datasets in E and SE Australia.

The performed processing and analysis was carried out competently, the obtained results appear to be mainly solid, and the manuscript is overall well written. However, I have large reservations about the study's significance. It basically just provides a few (way fewer than is apparent at first glance, see comments below) new data points that do not really show us anything new, and it also does not attempt to gain new insights by combining the obtained information with other existing evidence in a meaningful and

potentially novel way. I thus think that the study should not be published in its current shape, but the authors should be encouraged to submit an extended and improved version of the study that tries to, at least, improve on point 2 of my General comments (see below).

General comments:

As mentioned above, my main concern with this paper is its lack of significance. This can be broken down into two problems:

1. Data paucity and lack of novel results

The abstract talks about receiver functions from 32 stations (24 temporary from the BASS deployment, 8 permanent), but H-K results are only supplied for 13 stations (7 temporary, 6 permanent; the text says 14 stations but Table 2 only features 13), inversion results only for 6 (only permanent stations).

What the manuscript does not mention is that Moho depth estimates from receiver functions are already available from literature for 5 of the 6 permanent stations investigated (station CAN in Clitheroe et al., 2000; stations MOO, TOO, TAU, YNG in Ford et al., 2010; all 5 are also used in the AusMoho compilation of Kennett et al., 2011), and vp and vp/vs estimates for 4 of the 6 (Ford et al., 2010). This reduces the amount of new results to H-K stacking results from 7 temporary stations, and H-K stacking plus inversion for one permanent station (CNB) for which I could not find previous results.

This is a rather thin data base, and I think the authors should have mentioned the previous results I just listed, and discussed whether their new values agree or disagree with these previous findings (they are largely consistent, as far as I can see). Failure to do that appears to wrongly imply that all reported values are novel. Comparison is only undertaken with other previously analyzed stations in the region (Figure 9), which adds to this impression.

Lastly, it would make sense to compare the obtained crustal thicknesses to the

[Figure]

Australia-wide Moho model AusMoho (freely available from the webpage of ANU: http://rses.anu.edu.au/seismology/AuSREM/AusMoho/). This interpolated model offers predictions (interpolated values) for the positions of the newly analysed stations, thus it offers the possibility to check whether these results change or confirm the current state of knowledge (at first glance, they rather confirm).

2. Lack of constraints for interpretation

The authors offer a detailed introduction to models of crustal formation and geologic evolution of SE Australia in Sections 1 and 2. I am no expert in these things, but I get the impression that a thorough survey of the literature was performed.

The Discussion section then attempts to relate the rather poor data base (see above) to all kinds of geological processes that have been previously proposed. I think the authors are doing an OK job in relating their results to some published works, but the central problem is that the few newly obtained data are mostly interpreted in isolation. It would make a lot of sense to do cross plots with results from other geophysical studies, trying to see more by combining different datasets. This is not done at all, which is even more surprising considering that the first author published two other seismological studies on the same area, even partially using the same stations, last year (Bello et al., 2019a,b; teleseismic tomography and shear-wave splitting). While both of these methods rather illuminate deeper, sub-crustal structure, a combined interpretation would allow a much better discussion and potentially offer novel insights. I wonder why this is not done here, unless the authors want to publish this in yet another paper (which would be slicing it rather thinly). Looking at these previous papers, I also wonder why there was no attempt to use at least some of the huge number of WOMBAT temporary stations that was harvested for the teleseismic tomography in the present study, to derive some more (novel) data points.

In the following I will supply more minor comments by line number:

ll.24/25: I found it rather confusing that the authors talk about vp/vs ratio and Poisson

ratio separately (here and also in Section 7.2), although these properties are directly related (see Equation 2) and thus one does not offer additional information compared to the other.

ll.54-57: Some parts of a sentence are apparently missing here.

ll.93-100: If VanDieland is just a conceptual microcontental block in one of the models that is routinely used to explain the genesis of the region, why is the term used to reference station locations (e.g. Table 2). Shouldn't geographical regions that are independent of interpretation framework be used for this?

ll.102-154 (Section 3): I personally dislike this type of listing of existing studies, going study by study and explaining the methodology of each. This is unnecessarily bloated and in the end the reader doesn't take away much beyond "people have worked in this area before". It would be better to include the geophysical evidence into the presentation of evolution concepts given in Section 2 (and partly Section 1).

l.152: part of the sentence is missing (?)

ll.174/175: I disagree on this claim

l.185: "by using the clarity of the direct arrivals"; was this a purely visual selection or were there fixed criteria? Some more detail would be useful

l.208: Although the paper of Zhu and Kanamori (2000) is cited here, the used weighting scheme (0.6/0.3/0.1) is not the same as in that paper (0.7/0.2/0.1).

l.213: How robust is the use of standard deviations as uncertainty estimates when there are only 4-6 measurements (as is the case for 5 of the 13 stations, see Table 2)? This should at least be mentioned/discussed.

l.240: "Our strict criteria ...": what were these criteria? It would be worth explaining how this was done, especially since it leads to a reduction from 32 to 6 stations.

l.243: Why have a subchapter 5.3.1 if there is no 5.3.2?

l.294ff: Why is it not even mentioned that there is a huge difference in Moho depth between the two different applied analysis techniques (H-K and inversion) for all (3) stations in the Lachlan Fold Belt that were investigated with both methods (see Table 2)? These differences are around 10 km, thus very significant.

Figures:

Figures 6 and 9: I find the black-to-white color scale not to be a very good choice here, it is quite hard to see at a glance where e.g. the crust is thick and where thin. A different color scale may be more appropriate.

Figures 7 and 8: It would be useful to show where the Moho was picked in the shear-wave velocity models. Taking the values from Table 2, I actually disagree with the picks for stations YNG and CAN. In both of these cases, the clearer jump in vs is much shallower than what is listed in Table 2 (around 35 and 40 km instead of 48 and 49 km), which would also be much more consistent with H-K results

References:

Bello, M., N. Rawlinson, D.G. Cornwell, E. Crowder, M. Salmon, A.M. Reading (2019a), Structure of the crust and upper mantle beneath Bass Strait, southeast Australia, from teleseismic body wave tomography, Physics of the Earth and Planetary Interiors, 294, Article 106276.

Bello, M., D.G. Cornwell, N. Rawlinson, A.M. Reading (2019b), Insights into the structure and dynamics of the upper mantle beneath Bass Strait, southeast Australia, using shear wave splitting, Physics of the Earth and Planetary Interiors, 289, 45-62.

Clitheroe, G., O. Gudmundsson, B.L.N. Kennett (2000), The crustal thickness of Australia, J. Geophys. Res., 105( B6), 13697– 13713, doi:10.1029/1999JB900317.

Ford, H.A., K.M. Fischer, D.L. Abt, C.A. Rychert, L.T. Elkins-Tanton (2010), The lithosphere–asthenosphere boundary and cratonic lithospheric layering beneath Australia from Sp wave imaging, Earth and Planetary Science Letters, 300 (3–4), 299-310.

Kennett, B.L.N., M. Salmon, E. Saygin, AusMoho Working Group (2011), AusMoho: the variation of Moho depth in Australia, Geophysical Journal International, 187(2), 946–958, https://doi.org/10.1111/j.1365-246X.2011.05194.x

Zhu, L., H. Kanamori (2000), Moho depth variation in southern California from teleseismic receiver functions, J. Geophys. Res., 105( B2), 2969– 2980, doi:10.1029/1999JB900322.

---

## Referee Comment (RC2) · Anonymous Referee #2 · 9 Aug 2020

Review of

Crustal structure of southeast Australia from teleseismic receiver functions by

BELLO, CORNWELL, RAWLINSON, READING, LIKKASON
* * *
This carefully researched and well-written contribution places solid constraints on the crustal structure of southeast Australia by the construction and inversion of teleseismic receiver functions underneath a series of high-quality seismic stations. Building on these results for the thickness and sharpness of the crust, the authors put forward a tectonic interpretation, or rather a substantiation of earlier geological theories, involving magmatic underplating, which places the structure of the region into a proper geodynamic context.

I have relatively little to offer in the form of scientific criticism or comments on the seismological methods, which are sound, well-established, and well executed, although I am making a number of suggestions related to the presentation of the materials.

I am judging the paper primarily on its seismological merits, and not on the finer points of the interpretation. My main point related to the interpretation is that the comparison with earlier results by other authors is mostly qualitative, in the form of a color-coded figure, where I would have preferred a more detailed cross-comparison including a statistical analysis of uncertainty. How different can two crustal models made at two nearby stations be before tension develops with the interpretation? How different can two crustal models made at the same station be before we must dig into the details in order to interpret one of them as "better", or both of them as "equivalent"? The authors leave a bit of material on the table here.

I am attaching a hand-annotated manuscript. I will number and restate my most important comments here. I will not repeat "obvious" but necessary corrections here.

MAJOR COMMENTS

L261 What are those degrees of freedom, how do you determine them? The reference to Gouveia and Scales is too vague.

L310 In the same vain. I know it is hard to formally justify, but if you have the right number of degrees of freedom, and you have the right amount of independence in the entries of the summand, the reduced-chi-squared value that you should be aiming for is 1. Are you looking at the distribution of your misfits to establish that they ARE indeed chi-squared distributed? Are you sure that you are using the right amount of degrees of freedom? Are you sure that your lowest chi-squared values are not overly optimistic

(as in: that they could be nearly perfect fits to models with too many free parameters).

L832 I assume we are talking about the same criterion here, and so the caption should explicitly refer to it.

On the whole, I would like to read more about your misfit criterion, and I would like you to make explicit the now implicit distributional assumptions made about your metric.

L831 I definitely would put the numbers in call-out boxes on the maps also. A color scale is hard to read for some, and any additional clarity that can be gleaned from a multiplicity of representation is to be welcomed.

L835 Let the caption teach us how to read the top and bottom axes in the left-hand panel.

L850 Again, it is hard to see differences when they are presented on a busy colored map in a smooth gray-scale representation. A table would be nice in the main text. Spell out the differences, attempt to make sense of them relative to their uncertainty and their spatial proximity. Make us confident that your study is not just "another opinion", make us confident that other studies weren't just "another study", in other words: integrate the results of your an other studies and talk us through the similarities and differences. In the text, emphasize the common points and the differences, in particular in light of the interpretation.

MINOR COMMENTS

L10, L15, L26 "understanding", "this", "explains" -> those are all vague terms. After reading the manuscript it became clear to me that you had more detail in mind, some of which you have room to put into the abstract.

L17 "postulated Precambrian continental" -> I propose "putative" if the postulate refers to the fragment being "continental" or "putatively" if it refers to being "Precambrian".

L50, L55 -> Establish a consistent notation and typographical conventions

L150, L152, L186, L186 -> Fix typos and inconsistencies

L287 "relatively average to high" -> we need a basis for comparison, and a different word than "average" - in my book, values are not "average" unless they are "averages", and you most likely mean that these values are "unremarkable", "usually/frequently observed" (compared to what then?)

L325 -> Fix typo/inconsistency

L376 There is a lack of referencing in this sentence, which must refer to specific studies for each of the assertions made in it. Also "depicted" is not the greatest choice of word here.

L447 -> Fix typo/inconsistency

L777, L788, L791, L809 -> Fix capitalization

L798 Personally I would leave ETOPO1 out of the caption unless I was willing to put a color scale to it. At this scale and with this projection and without a color scale it's immaterial what topography model is being used.

L802 I would label the phases with letters on the graph also, right now the colors are not all that distinct on the screen, and they won't be on a black and white printer or photocopier, either.

END

Please also note the supplement to this comment:
https://se.copernicus.org/preprints/se-2020-74/se-2020-74-RC2-supplement.pdf

**Supplement:**

[revised manuscript text omitted]

↳ this is not useful unless there is a topography scale in meter, and you want us to read it.

[Figure]

[Figure]

**Figure 4: Stacked receiver functions from Australian National Seismic Network (ANSN) stations TOO, YNG, MOO**
**and GSN station TAU. Small arrows indicate arrival of the *Ps* (black), *PpPs* (red) and *PpPs + PsPs* (blue) phases from**
**the Moho.**

[Figure]

[Figure]

[Figure]

Figure 5: Results from the $H$-$\kappa$ stacking analysis for RFs (Zhu and Kanamori, 2000) at stations MOO, CAN and
TOO. In each case (a) Normalised amplitudes of the stack over all back-azimuths along the travel time curves
corresponding to the $P_s$ and $P_pP_s$ phases. (b) Corresponding stacked receiver function for each station.

[Figure]

[Figure]

[Figure]

Figure 6: (a) Variations in crustal thickness and (b) $V_p/V_s$ ratio taken from the linear ($H$-$\kappa$) stacking results (Table 2).
Crustal thickness varies between $30.5 \pm 0.1$ km and $39.1 \pm 0.5$ km. Thinner crust in Bass Strait can be seen flanked by
a relatively thicker crust to the north and south. $V_p/V_s$ ratios vary from $1.65 \pm 0.02$ to $1.75 \pm 0.02$. Thick black dashed
line denotes the boundary of VanDieland. Thick red dashed line outlines the boundary of East Tasmania Terrane
and eastern Bass Strait (ETT+EB). Thick white dashed line highlights the eastern part of the Lachlan Fold Belt.
Topography/bathymetry is based on the Etopo1 dataset (Amante and Eakins, 2009).

[Figure]

[Figure]

**Figure 7: (Left)** Seismic velocity models for CAN and MOO stations obtained from the neighbourhood algorithm
(Sambridge 1999a). The grey area indicates all the models searched by the algorithm. The best 1000 models are
indicated by the yellow to green colours; the best one (smallest misfit) corresponds to the red line, both for *S*-wave
velocity and $V_p/V_s$ ratio and the white line is the average velocity model. **(Right)** Waveform matches between the
observed stacked receiver functions (black) and predictions (blue) based on the best models.

*describe how to read the double axes for the top and the bottom. NOT labeling the 2.5 and 3.0. Vp/vs ratio would send a clear message that the leftmost pair of curves are the ons to which the lower axis applies*

*what measure, and units of misfit, are you using here (if it is the $\chi^2$ that you discuss in the text, you have to state that and stick with it)*

[Figure]

[Figure]

Figure 8: (Left) Seismic velocity models for stations TOO and YNG obtained from the neighbourhood algorithm.
(Right) Comparison between the observed stacked and the predicted receiver functions from the NA inversion. See
Figure 7 caption for more details.

[Figure]

[Figure]

[Figure]

Figure 9: (a) Map showing crustal thickness variations based on the S-wave velocity inversion results of this study
(stars) and previous studies (octagons) (Fontaine et al., 2013; Shibitani, 1996; Collins, 1991) and (b) comparison of
crustal thickness variations based on the *H-κ* grid search results of this study (stars) and previous results from the
study of Tkalčić et al. (2012) (octagons). Topography/bathymetry is based on the Etopo1 dataset (Amante and
Eakins, 2009).

*I can't make much of the gradations of colors. While the numbers, give us a table, let us appreciate the difference*

---

## Author Comment (AC1) · 30 Nov 2020

While a response to comments is provided as plain text below, we recommend that the full response document, including marked-up revised manuscript and figures, be accessed via the supplementary PDF file provided. It is easier to navigate.

Comment: In the present manuscript, the authors calculated and evaluated Ps receiver functions at a total of 32 seismic stations in southeast Australia, most of them situated along both sides of Bass Strait (Victoria and Tasmania). They applied H-K stacking and receiver function inversion using the neighborhood algorithm, relatively standard techniques that have already been applied to different datasets in E and SE Australia.

The performed processing and analysis was carried out competently, the obtained re-

sults appear to be mainly solid, and the manuscript is overall well written. However, I have large reservations about the study's significance. It basically just provides a few (way fewer than is apparent at first glance, see comments below) new data points that do not really show us anything new, and it also does not attempt to gain new insights by combining the obtained information with other existing evidence in a meaningful and potentially novel way. I thus think that the study should not be published in its current shape, but the authors should be encouraged to submit an extended and improved version of the study that tries to, at least, improve on point 2 of my General comments (see below).

Response: We thank the reviewer for their constructive criticism of the manuscript. We acknowledge that the receiver function results we obtain only represent a portion of the stations for which data are available, but we were very careful to remove poor quality data, and this resulted in fairly substantial culling of the dataset, which is inherently noisy due to the proximity of most stations to Bass Strait and the Southern Ocean. An extensive re-examination of the data has been made in response to this point, including re-processing the receiver functions and a less automated assessment. As a result. we have been able to add an additional four H-$\kappa$ stacking results for the portable network (stations BA03, BA09, BA19 and BA20) and eight NA inversion results for the portable network (BA02, BA07, BA08, BA09, BA13, BA17, BA19 and BA20) – see the revised Table 2. These new results make a substantial contribution to the revised paper, and has meant that the Abstract, Discussion and Conclusion sections have been substantially modified. We also take the point regarding the exploitation of existing evidence to improve our interpretation, and have put considerable effort towards achieving this in the revised manuscript.

Comment: As mentioned above, my main concern with this paper is its lack of significance. This can be broken down into two problems:

1. Data paucity and lack of novel results The abstract talks about receiver functions from 32 stations (24 temporary from the BASS deployment, 8 permanent), but H-K

results are only supplied for 13 stations(7 temporary, 6 permanent; the text says 14 stations but Table 2 only features 13), inversion results only for 6 (only permanent stations).

What the manuscript does not mention is that Moho depth estimates from receiver functions are already available from literature for 5 of the 6 permanent stations investigated (station CAN in Clitheroe et al., 2000; stations MOO, TOO, TAU, YNG in Ford etal., 2010; all 5 are also used in the AusMoho compilation of Kennett et al., 2011), and vp and vp/vs estimates for 4 of the 6 (Ford et al., 2010). This reduces the amount of new results to H-K stacking results from 7 temporary stations, and H-K stacking plus inversion for one permanent station (CNB) for which I could not find previous results.

This is a rather thin data base, and I think the authors should have mentioned the previous results I just listed, and discussed whether their new values agree or disagree with these previous findings (they are largely consistent, as far as I can see). Failure to do that appears to wrongly imply that all reported values are novel. Comparison is only undertaken with other previously analyzed stations in the region (Figure 9), which adds to this impression. Lastly, it would make sense to compare the obtained crustal thicknesses to the Australia-wide Moho model AusMoho (freely available from the webpage of ANU:http://rses.anu.edu.au/seismology/AuSREM/AusMoho/). This interpolated model offers predictions (interpolated values) for the positions of the newly analysed stations, thus it offers the possibility to check whether these results change or confirm the cur-rent state of knowledge (at first glance, they rather confirm).

Response: We acknowledge that what we wrote could be interpreted as a claim that we obtain receiver functions from all 32 stations, so have added a subsequent sentence that clarifies the actual number of usable receiver functions we extract (see lines 13-15). As noted above, we have re-assessed the temporary station data, and applied different receiver function assessment criteria, which has yielded additional results that we hope will help allay the reviewer's concern regarding data paucity.

In response to the second point, we now make mention of the previous Moho depth estimates for five of the six permanent stations, and show that they are consistent with our results. See lines 398-407.

In response to the last point, we have added a new figure (Figure 11) which compares all of our Moho depth results with AusREM estimates. As the reviewer notes, they are in general quite consistent, but with a few exceptions that we discuss in a new section (7.3) in the manuscript – see lines 560-597.

Comment: 2. Lack of constraints for interpretation The authors offer a detailed introduction to models of crustal formation and geologic evolution of SE Australia in Sections 1 and 2. I am no expert in these things, but I get the impression that a thorough survey of the literature was performed. The Discussion section then attempts to relate the rather poor data base (see above) to all kinds of geological processes that have been previously proposed. I think the authors are doing an OK job in relating their results to some published works, but the central problem is that the few newly obtained data are mostly interpreted in isolation. It would make a lot of sense to do cross plots with results from other geophysical studies, trying to see more by combining different datasets. This is not done at all, which is even more surprising considering that the first author published two other seismological studies on the same area, even partially using the same stations, last year (Bello et al.,2019a,b; teleseismic tomography and shear-wave splitting). While both of these methods rather illuminate deeper, sub-crustal structure, a combined interpretation would allow a much better discussion and potentially offer novel insights. I wonder why this is not done here, unless the authors want to publish this in yet another paper (which would be slicing it rather thinly). Looking at these previous papers, I also wonder why there was no attempt to use at least some of the huge number of WOMBAT temporary stations that was harvested for the teleseismic tomography in the present study, to derive some more (novel) data points.

Response: The reviewer makes a fair point with regard to the lack of comparison with the results from other relevant studies. As a consequence, we have made the following

changes: 1. As noted above, we now compare our results with the AuSMoho model (Figure 11 and Section 7.3). 2. We also compare our results in Tasmania to the study of Rawlinson et al. (2001) who invert wide-angle traveltimes (both reflection and refraction) for crustal velocity and Moho geometry (see Supplementary Figure S9). 3. We also include a new figure, which provides a joint interpretation of our results from teleseismic tomography, shear wave splitting and receiver function analysis (see Figure 12) and discuss it in detail in a new section (7.4) – see lines 598-623.

With regard to the last point, the WOMBAT stations were all short period stations (1 Hz corner frequency), many were only vertical component, and deployment times were often less than a year. All the Victorian stations and those deployed in northern Tasmania were vertical component, which means that it was not possible to extract receiver functions from them.

Comment: ll.24/25: I found it rather confusing that the authors talk about vp/vs ratio and Poisson ratio separately (here and also in Section 7.2), although these properties are directly related (see Equation 2) and thus one does not offer additional information compared to the other.

Response: We agree with the reviewer's comment, and have revised the manuscript so that it just refers to Vp/Vs ratio (see Abstract and Section 7.2).

Comment: ll.54-57: Some parts of a sentence are apparently missing here.

Response: As far as we can tell, the sentence in question appears to be complete. However, we have broken it up into two sentences, which hopefully improves its clarity (see lines 67-71).

Comment: ll.93-100: If VanDieland is just a conceptual microcontental block in one of the models that is routinely used to explain the genesis of the region, why is the term used to reference station locations (e.g. Table 2). Shouldn't geographical regions that are independent of interpretation framework be used for this?

Response: This is a fair point, and we have now removed such references (e.g. Table 2).

Comment: ll.102-154 (Section 3): I personally dislike this type of listing of existing studies, going study by study and explaining the methodology of each. This is unnecessarily bloated and in the end the reader doesn't take away much beyond "people have worked in this area before". It would be better to include the geophysical evidence into the presentation of evolution concepts given in Section 2 (and partly Section 1).

Response: We think it is worthwhile to summarise the results of previous geophysical studies, so have retained this section. However, we do acknowledge the reviewer's point, and therefore have changed it considerably to focus more on the outcomes of these studies. This includes largely removing the first paragraph, which merely lists the range of techniques that have been applied and relevant references. See lines 116-185 of the revised manuscript.

Comment: l.152: part of the sentence is missing (?)

Response: This sentence appears to be complete as far as we can tell.

Comment: ll.174/175: I disagree on this claim

Response: We have deleted this sentence (see lines 221-222).

Comment: l.185: "by using the clarity of the direct arrivals"; was this a purely visual selection or were there fixed criteria? Some more detail would be useful.

Response: We have added "visual clarity" to the sentence in question (see line 240).

Comment: l.208: Although the paper of Zhu and Kanamori (2000) is cited here, the used weighting scheme (0.6/0.3/0.1) is not the same as in that paper (0.7/0.2/0.1).

Response: We have removed the citation (see line 270-271).

Comment: l.213: How robust is the use of standard deviations as uncertainty estimates

when there are only 4-6 measurements (as is the case for 5 of the 13 stations, see Table 2)? This should at least be mentioned/discussed.

Response: This is a fair point, and we have added an additional sentence to explain the limitations of this approach (see lines 276-278).

Comment: l.240: "Our strict criteria ...": what were these criteria? It would be worth explaining how this was done, especially since it leads to a reduction from 32 to 6 stations.

Response: We have rephrased this sentence to make it clear that we are using visual criteria for acceptance (see line 305-306).

Comment: l.243: Why have a subchapter 5.3.1 if there is no 5.3.2?

Response: We have removed this heading (see line 308).

Comment: l.294ff: Why is it not even mentioned that there is a huge difference in Moho depth between the two different applied analysis techniques (H-K and inversion) for all (3) stations in the Lachlan Fold Belt that were investigated with both methods (see Table2)? These differences are around 10 km, thus very significant.

Response: We have revisited these Moho picks from the NA inversion, and determined that they were not done correctly. As can be seen in Table 2, they have been changed, with YNG H-$\kappa$ depth=37 km, NA depth=35km; CAN H-$\kappa$ depth=39 km, NA depth=40 km; CNB H$\kappa$ - depth=38 km, NA depth=39 km. However, it is worth noting that previous RF inversion results have favoured a Moho that is $\sim$10 km deeper beneath this region. We have discussed this in Section 7.1 – see lines 432-440.

Comment: Figures 6 and 9: I find the black-to-white color scale not to be a very good choice here, it is quite hard to see at a glance where e.g. the crust is thick and where thin. A different color scale may be more appropriate.

Response: These figures have been redone using a colour scale that makes it easier to distinguish between thick and thin crust - see Figures 6 and 10.

Comment: Figures 7 and 8: It would be useful to show where the Moho was picked in the shear-wave velocity models. Taking the values from Table 2, I actually disagree with the picks for stations YNG and CAN. In both of these cases, the clearer jump in vs is much shallower than what is listed in Table 2 (around 35 and 40 km instead of 48 and 49km), which would also be much more consistent with H-K results.

Response: The pick locations have now been included in Figures 7-9, noting that we also include two example RF inversions from the temporary array in the new Figure 9. As noted in a previous comment, the original picks for YHG, CAN and CNB were too deep, and have been revised, so the reviewer is correct. However, please also refer to lines 432-440 of the manuscript for an explanation of why the precise Moho depth might be difficult to estimate here.

Please also note the supplement to this comment:
https://se.copernicus.org/preprints/se-2020-74/se-2020-74-AC1-supplement.pdf

**Supplement:**

**Response to reviewer comments on "Crustal structure of southeast Australia from teleseismic receiver functions" (manuscript se-2020-74) by Bello et al.**

We thank the two reviewers for their constructive comments. The revision process has been delayed somewhat due to the lead author being unable to contribute due to health reasons. However, the co-authors feel that the updated version of the manuscript does address the primary comments and concerns of both reviewers, and is a distinct improvement on the original.

**Note:** Reviewer comments are in Blue text, and author responses are in black text. Line numbers refer to the marked up version of the manuscript.

**Reviewer #1**

**Comment:** In the present manuscript, the authors calculated and evaluated Ps receiver functions at a total of 32 seismic stations in southeast Australia, most of them situated along both sides of Bass Strait (Victoria and Tasmania). They applied H-K stacking and receiver function inversion using the neighborhood algorithm, relatively standard techniques that have already been applied to different datasets in E and SE Australia.

The performed processing and analysis was carried out competently, the obtained results appear to be mainly solid, and the manuscript is overall well written. However, I have large reservations about the study's significance. It basically just provides a few (way fewer than is apparent at first glance, see comments below) new data points that do not really show us anything new, and it also does not attempt to gain new insights by combining the obtained information with other existing evidence in a meaningful and potentially novel way. I thus think that the study should not be published in its current shape, but the authors should be encouraged to submit an extended and improved version of the study that tries to, at least, improve on point 2 of my General comments (see below).

**Response:** We thank the reviewer for their constructive criticism of the manuscript. We acknowledge that the receiver function results we obtain only represent a portion of the stations for which data are available, but we were very careful to remove poor quality data, and this resulted in fairly substantial culling of the dataset, which is inherently noisy due to the proximity of most stations to Bass Strait and the Southern Ocean. An extensive re-examination of the data has been made in response to this point, including re-processing the receiver functions and a less automated assessment. As a result. we have been able to add an additional four H-κ stacking results for the portable network (stations BA03, BA09, BA19 and BA20) and eight NA inversion results for the portable network (BA02, BA07, BA08, BA09, BA13, BA17, BA19 and BA20) – see the revised Table 2. These new results make a substantial contribution to the revised paper, and has meant that the Abstract, Discussion and Conclusion sections have been substantially modified. We also take the point regarding the exploitation of existing evidence to improve our interpretation, and have put considerable effort towards achieving this in the revised manuscript.

**Comment:** As mentioned above, my main concern with this paper is its lack of significance. This can be broken down into two problems:

1. Data paucity and lack of novel results
The abstract talks about receiver functions from 32 stations (24 temporary from the BASS deployment, 8 permanent), but H-K results are only supplied for 13 stations(7 temporary, 6 permanent; the text says 14 stations but Table 2 only features 13), inversion results only for 6 (only permanent stations).

What the manuscript does not mention is that Moho depth estimates from receiver functions are already available from literature for 5 of the 6 permanent stations investigated (station CAN in Clitheroe et al., 2000; stations MOO, TOO, TAU, YNG in Ford etal., 2010; all 5 are also used in the AusMoho compilation of Kennett et al., 2011), and vp and vp/vs estimates for 4 of the 6 (Ford et al., 2010). This reduces the amount of new results to H-K stacking results from 7 temporary stations, and H-K stacking plus inversion for one permanent station (CNB) for which I could not find previous results.

This is a rather thin data base, and I think the authors should have mentioned the previous results I just listed, and discussed whether their new values agree or disagree with these previous findings (they are largely consistent, as far as I can see). Failure to do that appears to wrongly imply that all reported values are novel. Comparison is only undertaken with other previously analyzed stations in the region (Figure 9), which adds to this impression. Lastly, it would make sense to compare the obtained crustal thicknesses to the Australia-wide Moho model AusMoho (freely available from the webpage of ANU:http://rses.anu.edu.au/seismology/AuSREM/AusMoho/). This interpolated model offers predictions (interpolated values) for the positions of the newly analysed stations, thus it offers the possibility to check whether these results change or confirm the cur-rent state of knowledge (at first glance, they rather confirm).

**Response:** We acknowledge that what we wrote could be interpreted as a claim that we obtain receiver functions from all 32 stations, so have added a subsequent sentence that clarifies the actual number of usable receiver functions we extract (see lines 13-15). As noted above, we have re-assessed the temporary station data, and applied different receiver function assessment criteria, which has yielded additional results that we hope will help allay the reviewer's concern regarding data paucity.

In response to the second point, we now make mention of the previous Moho depth estimates for five of the six permanent stations, and show that they are consistent with our results. See lines 398-407.

In response to the last point, we have added a new figure (Figure 11) which compares all of our Moho depth results with AusREM estimates. As the reviewer notes, they are in general quite consistent, but with a few exceptions that we discuss in a new section (7.3) in the manuscript – see lines 560-597.

**Comment:** 2. Lack of constraints for interpretation
The authors offer a detailed introduction to models of crustal formation and geologic evolution of SE Australia in Sections 1 and 2. I am no expert in these things, but I get the impression that a thorough survey of the literature was performed. The Discussion section then attempts to relate the rather poor data base (see above) to all kinds of geological processes that have been previously proposed. I think the authors are doing an OK job in relating their results to some published works, but the central problem is that the few newly obtained data are mostly interpreted in isolation. It would make a lot of sense to do cross plots with results from other geophysical studies, trying to see more by combining different datasets. This is not done at all, which is even more surprising considering that the first author published two other seismological studies on the same area, even partially using the same stations, last year (Bello et al.,2019a,b; teleseismic tomography and shear-wave splitting). While both of these methods rather illuminate deeper, sub-crustal structure, a combined interpretation would allow a much better discussion and potentially offer novel insights. I wonder why this is not done here, unless the authors want to publish this in yet another paper (which would be slicing it rather thinly). Looking at these previous papers, I also wonder why there was no attempt to use at least some of the huge number of WOMBAT temporary stations that was harvested for the teleseismic tomography in the present study, to derive some more (novel) data points.

**Response:** The reviewer makes a fair point with regard to the lack of comparison with the results from other relevant studies. As a consequence, we have made the following changes:

1. As noted above, we now compare our results with the AuSMoho model (Figure 11 and Section 7.3).
2. We also compare our results in Tasmania to the study of Rawlinson et al. (2001) who invert wide-angle traveltimes (both reflection and refraction) for crustal velocity and Moho geometry (see Supplementary Figure S9).
3. We also include a new figure, which provides a joint interpretation of our results from teleseismic tomography, shear wave splitting and receiver function analysis (see Figure 12) and discuss it in detail in a new section (7.4) – see lines 598-623.

With regard to the last point, the WOMBAT stations were all short period stations (1 Hz corner frequency), many were only vertical component, and deployment times were often less than a year. All the Victorian stations and those deployed in northern Tasmania were vertical component, which means that it was not possible to extract receiver functions from them.

**Comment:** ll.24/25: I found it rather confusing that the authors talk about vp/vs ratio and Poisson ratio separately (here and also in Section 7.2), although these properties are directly related (see Equation 2) and thus one does not offer additional information compared to the other.

**Response:** We agree with the reviewer's comment, and have revised the manuscript so that it just refers to *Vp/Vs* ratio (see Abstract and Section 7.2).

**Comment:** ll.54-57: Some parts of a sentence are apparently missing here.

**Response:** As far as we can tell, the sentence in question appears to be complete. However, we have broken it up into two sentences, which hopefully improves its clarity (see lines 67-71).

**Comment:** ll.93-100: If VanDieland is just a conceptual microcontental block in one of the models that is routinely used to explain the genesis of the region, why is the term used to reference station locations (e.g. Table 2). Shouldn't geographical regions that are independent of interpretation framework be used for this?

**Response:** This is a fair point, and we have now removed such references (e.g. Table 2).

**Comment:** ll.102-154 (Section 3): I personally dislike this type of listing of existing studies, going study by study and explaining the methodology of each. This is unnecessarily bloated and in the end the reader doesn't take away much beyond "people have worked in this area before". It would be better to include the geophysical evidence into the presentation of evolution concepts given in Section 2 (and partly Section 1).

**Response:** We think it is worthwhile to summarise the results of previous geophysical studies, so have retained this section. However, we do acknowledge the reviewer's point, and therefore have changed it considerably to focus more on the outcomes of these studies. This includes largely removing the first paragraph, which merely lists the range of techniques that have been applied and relevant references. See lines 116-185 of the revised manuscript.

**Comment:** l.152: part of the sentence is missing (?)

**Response:** This sentence appears to be complete as far as we can tell.

**Comment:** ll.174/175: I disagree on this claim

**Response:** We have deleted this sentence (see lines 221-222).

**Comment:** l.185: "by using the clarity of the direct arrivals"; was this a purely visual selection or were there fixed criteria? Some more detail would be useful.
**Response:** We have added "visual clarity" to the sentence in question (see line 240).

**Comment:** l.208: Although the paper of Zhu and Kanamori (2000) is cited here, the used weighting scheme (0.6/0.3/0.1) is not the same as in that paper (0.7/0.2/0.1).
**Response:** We have removed the citation (see line 270-271).

**Comment:** l.213: How robust is the use of standard deviations as uncertainty estimates when there are only 4-6 measurements (as is the case for 5 of the 13 stations, see Table 2)? This should at least be mentioned/discussed.
**Response:** This is a fair point, and we have added an additional sentence to explain the limitations of this approach (see lines 276-278).

**Comment:** l.240: "Our strict criteria ...": what were these criteria? It would be worth explaining how this was done, especially since it leads to a reduction from 32 to 6 stations.
**Response:** We have rephrased this sentence to make it clear that we are using visual criteria for acceptance (see line 305-306).

**Comment:** l.243: Why have a subchapter 5.3.1 if there is no 5.3.2?
**Response:** We have removed this heading (see line 308).

**Comment:** l.294ff: Why is it not even mentioned that there is a huge difference in Moho depth between the two different applied analysis techniques (H-K and inversion) for all (3) stations in the Lachlan Fold Belt that were investigated with both methods (see Table2)? These differences are around 10 km, thus very significant.
**Response:** We have revisited these Moho picks from the NA inversion, and determined that they were not done correctly. As can be seen in Table 2, they have been changed, with YNG H-κ depth=37 km, NA depth=35km; CAN H- κ depth=39 km, NA depth=40 km; CNB Hκ - depth=38 km, NA depth=39 km. However, it is worth noting that previous RF inversion results have favoured a Moho that is ~10 km deeper beneath this region. We have discussed this in Section 7.1 – see lines 432-440.

**Comment:** Figures 6 and 9: I find the black-to-white color scale not to be a very good choice here, it is quite hard to see at a glance where e.g. the crust is thick and where thin. A different color scale may be more appropriate.
**Response:** These figures have been redone using a colour scale that makes it easier to distinguish between thick and thin crust - see Figures 6 and 10.

**Comment:** Figures 7 and 8: It would be useful to show where the Moho was picked in the shear-wave velocity models. Taking the values from Table 2, I actually disagree with the picks for stations YNG and CAN. In both of these cases, the clearer jump in vs is much shallower than what is listed in Table 2 (around 35 and 40 km instead of 48 and 49km), which would also be much more consistent with H-K results.
**Response:** The pick locations have now been included in Figures 7-9, noting that we also include two example RF inversions from the temporary array in the new Figure 9. As noted in a previous comment, the original picks for YHG, CAN and CNB were too deep, and have been revised, so the reviewer is correct. However, please also refer to lines 432-440 of the manuscript for an explanation of why the precise Moho depth might be difficult to estimate here.

**Reviewer #2**

**Comment:** This carefully researched and well-written contribution places solid constraints on the crustal structure of southeast Australia by the construction and inversion of teleseismic receiver functions underneath a series of high-quality seismic stations. Building on these results for the thickness and sharpness of the crust, the authors put forward a tectonic interpretation, or rather a substantiation of earlier geological theories, involving magmatic underplating, which places the structure of the region into a proper geodynamic context.

I have relatively little to offer in the form of scientific criticism or comments on the seismological methods, which are sound, well-established, and well executed, although I am making a number of suggestions related to the presentation of the materials.

I am judging the paper primarily on its seismological merits, and not on the finer points of the interpretation. My main point related to the interpretation is that the comparison with earlier results by other authors is mostly qualitative, in the form of a color-coded figure, where I would have preferred a more detailed cross-comparison including a statistical analysis of uncertainty. How different can two crustal models made at two nearby stations be before tension develops with the interpretation? How different can two crustal models made at the same station be before we must dig into the details in order to interpret one of them as "better", or both of them as "equivalent"? The authors leave a bit of material on the table here.

I am attaching a hand-annotated manuscript. I will number and restate my most important comments here. I will not repeat "obvious" but necessary corrections here.

**Response:** We thank the reviewer for the positive comments, and in the revised manuscript, attempt to improve on the quantitative nature of the comparison with previous results.

**Comment:** L261 What are those degrees of freedom, how do you determine them? The reference to Gouveia and Scales is too vague.
**Response:** The degrees of freedom is equal to the number of observations minus the number of inversion parameters, which we now state in the manuscript (see lines 328-329).

**Comment:** L310 In the same vain. I know it is hard to formally justify, but if you have the right number of degrees of freedom, and you have the right amount of independence in the entries of the summand, the reduced-chi-squared value that you should be aiming for is 1. Are you looking at the distribution of your misfits to establish that they ARE indeed chi-squared distributed? Are you sure that you are using the right amount of degrees of freedom? Are you sure that your lowest chi-squared values are not overly optimistic (as in: that they could be nearly perfect fits to models with too many free parameters).
**Response:** This is a fair point, and ideally one would be aiming for a value of 1 to fit the data. However, apart from getting the number of degrees of freedom right, the noise estimate is also a factor, and its absolute value is poorly constrained. This may be why the chi-square values are on the low side, but we think it is reasonable to consider our measure of chi-square as a relative indicator of data fit, which we now acknowledge on lines 393-397. It is also worth noting that it is fairly typical of NA RF inversion studies to end up with chi-square misfit values well below 1 (e.g. Wu et al., 2015: Crustal shear wave velocity structure in the northeastern Tibet based on the Neighbourhood algorithm inversion of receiver functions. Geophysical Journal International, **212**, 1920-1931)

**Comment:** L832 I assume we are talking about the same criterion here, and so the caption should explicitly refer to it.

On the whole, I would like to read more about your misfit criterion, and I would like you to make explicit the now implicit distributional assumptions made about your metric.
**Response:** Yes, this is the same misfit measure, which is now clarified in the paper (see line 1014).

**Comment:** L831 I definitely would put the numbers in call-out boxes on the maps also. A color scale is hard to read for some, and any additional clarity that can be gleaned from a multiplicity of representation is to be welcomed.
**Response:** We now point out the depth values on the plots.

**Comment:** L835 Let the caption teach us how to read the top and bottom axes in the left-hand panel.
**Response:** We have changed the caption as requested (see line 1012).

**Comment:** L850 Again, it is hard to see differences when they are presented on a busy colored map in a smooth gray-scale representation. A table would be nice in the main text. Spell out the differences, attempt to make sense of them relative to their uncertainty and their spatial proximity. Make us confident that your study is not just "another opinion", make us confident that other studies weren't just "another study", in other words:integrate the results of your an other studies and talk us through the similarities and differences. In the text, emphasize the common points and the differences, in particular in light of the interpretation.
**Response:** We have changed the relevant figures (Figure 6 and 10) to make it easier to read the variations in Moho depth, and changed the discussion to make it more integrated, along the lines suggested by the reviewer. Table 2 also provides a quantitative summary of all our new results.

**Comment:** L10, L15, L26 "understanding", "this", "explains" -> those are all vague terms. After reading the manuscript it became clear to me that you had more detail in mind, some of which you have room to put into the abstract.
**Response:** We have made some modifications to the Abstract to improve upon clarity.

**Comment:** L17 "postulated Precambrian continental" -> I propose "putative" if the postulate refers to the fragment being "continental" or "putatively" if it refers to being "Precambrian".
**Response:** We have implemented this change (see line 21).

**Comment:** L50, L55 -> Establish a consistent notation and typographical conventions
**Response:** This has been done.

**Comment:** L150, L152, L186, L186 -> Fix typos and inconsistencies
**Response:** This has been corrected.

**Comment:** L287 "relatively average to high" -> we need a basis for comparison, and a different word than "average" - in my book, values are not "average" unless they are "averages",and you most likely mean that these values are "unremarkable", "usually/frequently observed" (compared to what then?
**Response:** The paragraph containing this text has been deleted as part of the revisions.

**Comment:** L325 -> Fix typo/inconsistency
**Response:** Typo has been corrected.

**Comment:** L376 There is a lack of referencing in this sentence, which must refer to specific studies for each of the assertions made in it. Also "depicted" is not the greatest choice of word here.

**Response:** We have replaced "depicted" with "revealed", and included a reference to the work of Christensen (1996). The reference to Owens and Zandt (1997) in the second point refers to the relationship between partial melt and Poisson's ratio. See lines 487-489.

**Comment:** L447 -> Fix typo/inconsistency
**Response:** Done.

**Comment:** L777, L788, L791, L809 -> Fix capitalization
**Response:** Done

**Comment:** L798 Personally I would leave ETOPO1 out of the caption unless I was willing to put a color scale to it. At this scale and with this projection and without a color scale it's immaterial what topography model is being used.
**Response:** We agree, but it is a journal requirement to state the source of any information we use in figures that was obtained from outside the current study.

**Comment:** L802 I would label the phases with letters on the graph also, right now the colors are not all that distinct on the screen, and they won't be on a black and white printer or photocopier, either.
**Response:** This suggestion has been implemented – see Figure 4.

**Annotated manuscript:** We also implemented the minor hand-annotated suggestions in the manuscript provided by Reviewer 2, which were mainly typos and other straight forward edits.

[revised manuscript text omitted]
 w̶e̶r̶e̶are separated into three groups (Fig. 2 ̶a̶n̶d̶ ̶T̶a̶b̶l̶e̶ ̶2̶) based on tectonic
setting̶s and the results obtained. Stations YNG, CAN, CNB, MILA and BA13 are located in the Lachlan Fold
Belt; stations BA02, BA11, BA19, BA20, TAU, MOO and TOO sit above the VanDieland microcontinental
block; and stations BA07, BA08, BA09 and BA17 lie in the East Tasmania Terrane and Eastern Bass Strait
(ETT+EB). Stations BA22 and BA24 lie to the west of VanDieland. This discussion focuses on crustal
thickness, ̶a̶n̶d̶ the nature of the Moho and crustal velocity and velocity ratio variations from *H-κ* stacking and
t̶h̶e̶ ̶n̶a̶t̶u̶r̶e̶ ̶o̶f̶ ̶t̶h̶e̶ ̶c̶r̶u̶s̶t̶ ̶f̶r̶o̶m̶ ̶*̶V̶p̶/̶V̶s̶*̶,̶ ̶P̶o̶i̶s̶s̶o̶n̶'̶s̶ ̶r̶a̶t̶i̶o̶ ̶a̶n̶d̶ the 1-D *S*-wave velocity models. Overall, the agreement
between Moho depths obtained from the H-κ stacking results and NA-inversion is generally within error (Table
2), which makes a joint interpretation m̶o̶r̶e̶ ̶s̶t̶r̶a̶i̶g̶h̶t̶ ̶f̶o̶r̶w̶a̶r̶d̶s̶more straight forward. Comparison is also made to
other studies that have examined crustal seismic properties in southeast Australia, and we attempt to integrate
our new findings with previous results from teleseismic tomography, SKS splitting and ambient noise
tomography in order to better understand the crust and upper mantle structure and dynamics beneath this region.

**7.1 Lateral variation of crustal thickness and nature of the Moho**

The RF analysis clearly reveals the presence of lateral changes in crustal thickness that span mainland Australia
through Bass Strait to Tasmania (Figures 6 and 10; in the latter case, RF depths from previous studies are also
included for reference). The stations located in the Palaeozoic Lachlan Fold Belt reveal̶s a generally thick crust
that ranges f̶r̶o̶m̶ ̶3̶6̶.̶5̶ ̶±̶ ̶4̶.̶4̶ ̶t̶o̶ ̶3̶9̶.̶1̶ ̶±̶ ̶0̶.̶5̶ ̶k̶m̶between ~37 and ~40 km. A̶t̶ ̶s̶t̶a̶t̶i̶o̶n̶ ̶C̶A̶N̶,̶ ̶t̶h̶e̶r̶e̶ ̶i̶s̶ ̶a̶ ̶d̶i̶s̶p̶a̶r̶i̶t̶y̶ ̶i̶n̶
c̶r̶u̶s̶t̶a̶l̶ ̶t̶h̶i̶c̶k̶n̶e̶s̶s̶ ̶o̶b̶t̶a̶i̶n̶e̶d̶ ̶b̶y̶ ̶t̶h̶e̶ ̶n̶o̶n̶-̶l̶i̶n̶e̶a̶r̶ ̶i̶n̶v̶e̶r̶s̶i̶o̶n̶ ̶m̶e̶t̶h̶o̶d̶ ̶(̶~̶4̶9̶ ̶k̶m̶)̶ ̶a̶n̶d̶ ̶*̶H̶-̶κ̶*̶ ̶s̶t̶a̶c̶k̶i̶n̶g̶ ̶t̶e̶c̶h̶n̶i̶q̶u̶e̶ ̶(̶3̶9̶.̶1̶ ̶±̶ ̶0̶.̶5̶
k̶m̶)̶.̶ ̶T̶h̶e̶ ̶r̶e̶a̶s̶o̶n̶ ̶a̶p̶p̶e̶a̶r̶s̶ ̶t̶o̶ ̶b̶e̶ ̶t̶h̶a̶t̶ ̶t̶h̶e̶ ̶*̶H̶-̶κ̶*̶ ̶s̶t̶a̶c̶k̶i̶n̶g̶ ̶a̶n̶a̶l̶y̶s̶i̶s̶ ̶a̶s̶s̶u̶m̶e̶s̶ ̶t̶h̶a̶t̶ ̶t̶h̶e̶ ̶c̶r̶u̶s̶t̶ ̶i̶s̶ ̶a̶ ̶s̶i̶n̶g̶l̶e̶ ̶l̶a̶y̶e̶r̶ ̶w̶i̶t̶h̶ ̶a̶
v̶e̶l̶o̶c̶i̶t̶y̶ ̶j̶u̶m̶p̶ ̶a̶c̶r̶o̶s̶s̶ ̶t̶h̶e̶ ̶M̶o̶h̶o̶,̶ ̶w̶h̶e̶r̶e̶a̶s̶ ̶t̶h̶e̶ ̶c̶r̶u̶s̶t̶-̶m̶a̶n̶t̶l̶e̶ ̶t̶r̶a̶n̶s̶i̶t̶i̶o̶n̶ ̶i̶s̶ ̶a̶c̶t̶u̶a̶l̶l̶y̶ ̶g̶r̶a̶d̶u̶a̶l̶;̶ ̶h̶e̶n̶c̶e̶ ̶i̶t̶ ̶i̶n̶s̶t̶e̶a̶d̶ ̶t̶a̶r̶g̶e̶t̶s̶ ̶a̶
s̶h̶a̶l̶l̶o̶w̶e̶r̶ ̶b̶o̶u̶n̶d̶a̶r̶y̶ ̶t̶h̶a̶t̶ ̶i̶s̶ ̶n̶o̶t̶ ̶t̶h̶e̶ ̶M̶o̶h̶o̶.̶ ̶T̶h̶e̶r̶e̶f̶o̶r̶e̶,̶ ̶t̶h̶e̶ ̶d̶e̶e̶p̶ ̶c̶r̶u̶s̶t̶a̶l̶ ̶s̶t̶r̶u̶c̶t̶u̶r̶e̶ ̶o̶b̶t̶a̶i̶n̶e̶d̶ ̶a̶t̶ ̶Y̶N̶G̶,̶ ̶C̶A̶N̶ ̶a̶n̶d̶ ̶C̶N̶B̶
i̶s̶ ̶p̶a̶r̶t̶ ̶o̶f̶ ̶a̶ ̶b̶r̶o̶a̶d̶ ̶v̶e̶l̶o̶c̶i̶t̶y̶ ̶t̶r̶a̶n̶s̶i̶t̶i̶o̶n̶ ̶z̶o̶n̶e̶ ̶f̶r̶o̶m̶ ̶c̶r̶u̶s̶t̶ ̶t̶o̶ ̶m̶a̶n̶t̶l̶e̶.̶ ̶T̶h̶e̶ ̶c̶r̶u̶s̶t̶a̶l̶ ̶t̶h̶i̶c̶k̶n̶e̶s̶s̶ ̶a̶n̶d̶ ̶M̶o̶h̶o̶ ̶t̶r̶a̶n̶s̶i̶t̶i̶o̶n̶ ̶z̶o̶n̶e̶
b̶e̶n̶e̶a̶t̶h̶ ̶t̶h̶e̶ ̶L̶a̶c̶h̶l̶a̶n̶ ̶O̶r̶o̶g̶e̶n̶ ̶o̶b̶t̶a̶i̶n̶e̶d̶ ̶b̶y̶ ̶t̶h̶e̶ ̶n̶o̶n̶l̶i̶n̶e̶a̶r̶ ̶i̶n̶v̶e̶r̶s̶i̶o̶n̶ ̶m̶e̶t̶h̶o̶d̶ ̶i̶s̶ ̶c̶o̶n̶s̶i̶s̶t̶e̶n̶t̶ ̶w̶i̶t̶h̶ ̶p̶r̶e̶v̶i̶o̶u̶s̶ ̶r̶e̶f̶r̶a̶c̶t̶i̶o̶n̶
a̶n̶d̶ ̶R̶F̶ ̶s̶t̶u̶d̶i̶e̶s̶ Although the Moho was picked as a velocity jump for stations YNG, CAN and CNB, the
velocity nonetheless tends to continue to increase with depth below the discontinuity. This, coupled with the fact
that Clitheroe et al. (2000) estimate the Moho to be almost 10 km deeper beneath CAN, is consistent with the
presence of mafic underplating (Shibutani et al., 1996; Clitheroe et al., 2000; Collins et al., 2003; Fontaine et al.,
2013a,b). T̶h̶e̶ ̶c̶r̶u̶s̶t̶a̶l̶ ̶t̶h̶i̶c̶k̶n̶e̶s̶s̶ ̶v̶a̶r̶i̶a̶t̶i̶o̶n̶s̶ ̶a̶n̶d̶ ̶l̶a̶c̶k̶ ̶o̶f̶ ̶a̶ ̶c̶l̶e̶a̶r̶ ̶M̶o̶h̶o̶ ̶a̶t̶ ̶t̶h̶e̶ ̶b̶a̶s̶e̶ ̶o̶f̶ ̶t̶h̶e̶ ̶L̶a̶c̶h̶l̶a̶n̶ ̶O̶r̶o̶g̶e̶n̶ ̶c̶r̶u̶s̶t̶ ̶m̶a̶y̶

[revised manuscript text omitted]

**1-D S-wave velocity inversion**

This section presents seismic S-wave velocity models for stations listed in Table 2, except those already shown in the main manuscript, obtained from receiver function inversion using the neighbourhood algorithm. The grey area indicates all the models searched by the algorithm. The best 1000 models are indicated in the yellow to green colour; the best model (smallest misfit) corresponds to the red line, both for S-wave velocity and *Vp/Vs* ratio, whereas the white line is the average velocity model computed from the best 1000 models. (Right) Waveform matches between the observed stacked receiver functions (black) and prediction (grey) based on the average of the best 1000 models.

[Figure]

**Figure S5:** (Left) Density plot of S-wave velocity models and (right) observed and synthetic RF plots for stations CNB and TAU

[Figure]

**Figure S6:** (Left) Density plot of S-wave velocity models and (right) observed and synthetic RF plots for stations BA07 and BA09.

[Figure]

**Figure S7:** (Left) Density plot of S-wave velocity models and (right) observed and synthetic RF plots for stations BA18 and BA20.

[Figure]

**Figure S8:** (Left) Density plot of S-wave velocity models and (right) observed and synthetic RF plots for stations BA07 and BA09.

[Figure]

**Figure S9:** Comparison between Moho map of Rawlinson et al. (2001) and receiver function results from this study. Small circles correspond to results from H-κ stacking, and large circles correspond to results from NA inversion.

---

## Author Response (AR2)

**Response to reviewer comments on "Crustal structure of southeast Australia from teleseismic receiver functions" (manuscript se-2020-74) by Bello et al.**

5th January 2021

We thank Reviewer #1 for the additional minor comments on our manuscript. Below, we include a response to each point that has been raised.

**Note:** Reviewer comments are in Blue text, and author responses are in black text. Line numbers refer to the marked up version of the manuscript.

**Reviewer #1**

**Comment:** l.20 and 22: Capitalization should be consistent (VanDieland vs. Vandieland)
**Response:** The correct term is "VanDieland". We have undertaken a string search of the entire manuscript to ensure that capitalisation is consistent throughout (e.g. line 22).

**Comment:** l.83: I have never seen the term ``turbidite thrust system", and I frankly do not understand what it is supposed to mean. As far as I can see, the cited paper (Gray and Foster, 2004) also does not use this term
**Response:** Turbidites are often found in deep water fold and thrust belts at convergent margins. In this case, we were referring to the incorporation of turbidites in a thrust system caused by the closure of a marginal basin. However, since this is not a key part of the Gray and Foster model, we have just deleted the term to avoid confusion (see line 83).

**Comment:** l.88: contain + s
**Response:** This has been corrected i.e. "contain" has been replaced with "contains" - see line 88.

**Comment:** l.107: employed instead of deployed?
**Response:** We have implemented this change (see line 107).

**Comment:** l.333: The sentence starting with ``Looking" should, as far as I can see, not be part of the itemized list any more
**Response:** This is correct. We have moved this sentence to the end of the paragraph below – see lines 343-345.

**Comment:** l.400: crustal thickness (-es)
**Response:** We have changed "thicknesses" to "thickness" as requested (see line 405).

**Comment:** l.406: since the relationship between Poisson's ratio and vp/vs is not discussed, this statement is confusing. Best reformulate to vp/vs.
**Response:** This statement has been reformulated in terms of Vp/Vs as suggested – see lines 410-411.

**Comment:** l.427: should be a lower case k in ka
**Response:** This has been corrected – see line 432.

**Comment:** l.428 and before: check for consistency in having ``vp/vs" italized or non-italized
**Response:** We have checked (and corrected where necessary) all occurrences of *Vp/Vs* to ensure that they are italic.

**Comment:** l.434/517/521: more Vandielands with lower case d..
**Response:** These have all been corrected.

**Comment:** l.439: should be southeast Australia I guess
**Response:** Yes, this was a typo, and "Asia" has been replaced with "Australia" - see line 445.

**Comment:** l.782: spelling of modified
**Response:** This has been corrected – see line 816.

**Comment:** Figure 9: Why are the RF traces for stations BA13 and BA17 cut off at 0s, whereas the ones for the permanent stations (Figures 7+8) are not? Is this simply a plotting artefact, or does it indicate differences in the processing? (same for the Supplementary Figures)
**Response:** Yes, we used a different processing strategy for the temporary stations. This is now explained on lines 295-298.

**Comment:** Figure 10: caption needs to be updated; it describes stars and octagons, but the figure shows circles and triangles. One reference is misspelled (Shibutani)
**Response:** We have made the necessary corrections, including to the spelling of Shibutani – see lines 876-877.

**Comment:** Figure 12: This is a nice composite figure. However, the color scale seems to indicate that all shown velocities (or velocity differences) are vp, while the shown crustal velocities are vs from ambient noise tomography, hence vs (if I understand correctly). This should be indicated, ideally with a second scale bar for these velocities.
**Response:** It is true that the crustal model derived from ambient noise is defined in terms of Vsv (vertical component data were exploited), but when it was used in conjunction with teleseismic P-wave arrival times, it was first converted to Vp, which is what is shown in this figure. The caption has been updated accordingly (see lines 892-895).

**Comment:** Figures S5-S8: can Moho picks be shown, just as in Figures 7-9?
**Response:** The Moho picks are now included in Figures S5-S8.

**Comment:** Figure S9: what are the pink stars and the numbers associated to them? This is not mentioned in the caption. Also, it is not mentioned what red triangles (L1 to L15) are. Please clarify!"
**Response:** This has been clarified in Figure S9 as requested.

---

## Author Response (AR4)

**Response to reviewer comments on "Crustal structure of southeast Australia from teleseismic receiver functions" (manuscript se-2020-74) by Bello et al.**

19[th] January 2021

This is the same rejoinder as was submitted on the 14[th] January. We have not made any changes to the manuscript subsequent to acceptance, other than to update Table 1, so that it is no longer embedded as an image.

**Note:** Line numbers refer to the marked up version of the manuscript.

**Editor**

**Comment:** thank you for addressing the comments of the reviewer, anyways the part of figure 9 where the observed+synthetic RF are plotted does not look nice enough to be shown in the main text as is. Please consider to show the RF (at least the observed) starting at -5 s; please change the lines 295-298 referring to the best-family rather than to the superior results achieved by using the shorter time window [since what has been cut is pre-signal noise]. In case you have not the possibility to show the RFs starting at -5s, please move the panels containing the RFs to the supplementary material, and move lines 295-298 in the caption of that figure.

**Response:** We have adopted the second option due to the difficulties in showing the receiver functions starting at -5s (figure generation is not decoupled from the NA RF software). Specifically, we have moved the waveform fits associated with Figure 9 to Supplementary Figure S7. For consistency with the other RF plots in the Supplementary information, we also include the S-wave models, but note the duplication in the caption. We have also made changes to the text where appropriate, namely:

- Moving lines 295-297 to the caption of the new Supplementary Figure S7. We have also included this statement in the captions of the other BASS station RFs for consistency (Supplementary Figures S6, S8 and S9).
- Updating references to the Supplementary Figures in the manuscript, since there is an extra supplementary figure added (Figure S7).
- Editing the Figure 9 caption, since it now only contains the S-wave models. It also includes a reference to the location of the associated waveform fits.